# Dissecting Chain-of-Thought: Compositionality through In-Context Filtering and Learning

**Yingcong Li**
University of California, Riverside
yli692@ucr.edu

**Kartik Sreenivasan**    **Angeliki Giannou**
University of Wisconsin-Madison
{ksreenivasa2,giannou}@wisc.edu

**Dimitris Papailiopoulos**
University of Wisconsin-Madison
dimitris@papail.io

**Samet Oymak**
University of Michigan & UC Riverside
oymak@umich.edu

## Abstract

Chain-of-thought (CoT) is a method that enables language models to handle complex reasoning tasks by decomposing them into simpler steps. Despite its success, the underlying mechanics of CoT are not yet fully understood. In an attempt to shed light on this, our study investigates the impact of CoT on the ability of transformers to in-context learn a simple to study, yet general family of compositional functions: multi-layer perceptrons (MLPs). In this setting, we find that the success of CoT can be attributed to breaking down in-context learning of a compositional function into two distinct phases: focusing on and filtering data related to each step of the composition and in-context learning the single-step composition function. Through both experimental and theoretical evidence, we demonstrate how CoT significantly reduces the sample complexity of in-context learning (ICL) and facilitates the learning of complex functions that non-CoT methods struggle with. Furthermore, we illustrate how transformers can transition from vanilla in-context learning to mastering a compositional function with CoT by simply incorporating additional layers that perform the necessary data-filtering for CoT via the attention mechanism. In addition to these test-time benefits, we show CoT helps accelerate pretraining by learning shortcuts to represent complex functions and filtering plays an important role in this process. These findings collectively provide insights into the mechanics of CoT, inviting further investigation of its role in complex reasoning tasks.

## 1 Introduction

The advent of transformers [Vaswani et al., 2017] has revolutionized natural language processing, paving the way for remarkable performance in a wide array of tasks. LLMs, such as GPTs [Brown et al., 2020], have demonstrated an unparalleled ability to capture and leverage vast amounts of data, thereby facilitating near human-level performance across a variety of language generation tasks. Despite this success, a deep understanding of their underlying mechanisms remains elusive.

Chain-of-thought prompting [Wei et al., 2022c] is an emergent ability of transformers where the model solves a complex problem [Wei et al., 2022b], by decomposing it into intermediate steps. Intuitively, this underlies the ability of general-purpose language models to accomplish previously-unseen complex tasks by leveraging more basic skills acquired during the pretraining phase. Compositional learning and CoT has enjoyed significant recent success in practical language modeling tasks spanning question answering, code generation, and mathematical reasoning [Perez et al., 2021, Imani et al., 2023, Yuan et al., 2023]. In this work, we attempt to demystify some of the mechanics underlying this success and the benefits of CoT in terms of sample complexity and approximation power. To

37th Conference on Neural Information Processing Systems (NeurIPS 2023).

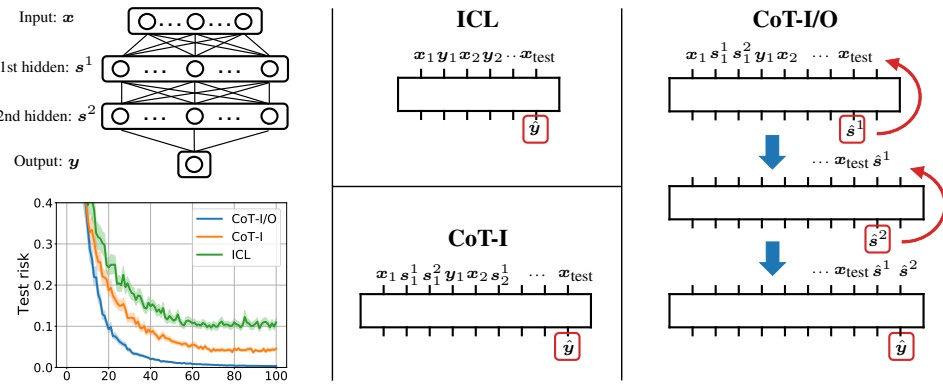

Figure 1: An illustration of ICL, CoT-I and CoT-I/O methods, using a 3-layer MLP as an example (top left, where $x$, $y$, $s^1$, $s^2$ denote input, output and hidden features respectivaly). The ICL method utilizes in-context examples in the form of $(x, y)$ and makes predictions directly based on the provided $x_{\text{test}}$. Both CoT-I and CoT-I/O methods admit prompts with samples formed by $(x, s^1, s^2, y)$. However, CoT-I/O uniquely makes recurrent predictions by re-inputting the intermediate output (as shown on the right). The performance of these methods is shown on the bottom left, with a more detailed discussion available in Section 4.

do this we explore the role of CoT in learning multi-layer perceptrons (MLPs) in-context, which we believe can lead to a first set of insightful observations. Throughout, we ask:

*Does CoT improve in-context learning of MLPs, and what are the underlying mechanics?*

In this work, we identify and thoroughly compare three schemes as illustrated in Figure 1. (a) ICL: In-context learning from input-output pairs provided in the prompt, (b) CoT-I: Examples in the prompt are augmented with intermediate steps, (c) CoT-I/O: The model also outputs intermediate steps during prediction. Our main contributions are:

- **Decomposing CoT into filtering and ICL:** As our central contribution, we establish a rigorous and experimentally-supported abstraction that decouples CoT prompting into a *filtering phase* and an *in-context learning (ICL) phase*. In *filtering*, the model attends to the relevant tokens within the prompt based on an instruction. In *ICL*, the model runs inference on the filtered prompt to output a *step* and then moves onto the next *step* in the chain. Our Theorem 1 develops a theoretical understanding of this two-step procedure and formalizes how filtering and ICL phases of CoT can be implemented via the self-attention mechanism to learn MLPs.
- **Approximation and sample complexity:** Through experiments and theory, we establish that intermediate steps in CoT-I improves the sample complexity of learning whereas step-by-step output improves the approximation ability through looping. Specifically, CoT-I/O can learn an MLP with input dimension $d$ and $k$ neurons using $\mathcal{O}(\max(k, d))$ in-context samples by filtering individual layers and solving them via linear regression – in contrast to the $\Omega(kd)$ lower bound without step-augmented prompt. As predicted by our theory, our experiments (see Sec. 4) identify a striking universality phenomenon (as $k$ varies) and also demonstrate clear approximation benefits of CoT compared to vanilla ICL.
- **Accelerated pretraining via learning shortcuts:** We construct deep linear MLPs where each layer is chosen from a discrete set of matrices. This is in contrast to the above setting, where MLP weights can be arbitrary. We show that CoT can dramatically accelerate pretraining by memorizing these discrete matrices and can infer all layers correctly from a *single* demonstration. Notably the pretraining loss goes to zero step-by-step where each step *"learns to filter a layer"*. Together, these showcase how CoT identifies composable shortcuts to avoid the need for solving linear regression. In contrast, we show that ICL (without CoT) collapses to linear regression performance as it fails to memorize exponentially many candidates (due to lack of composition).

The paper is organized as follows. In Section 2, we introduce the problem setup. Section 3 states our main theoretical results which decouple CoT into filtering and ICL. Section 4 provides empirical investigations of CoT with 2-layer MLPs, which validates our theoretical findings. Finally, we

elucidate the benefits of CoT during pretraining via experiments on deep linear MLPs in Section 5. Related work and discussion are provided in Sections 6 and 7, respectively.

## 2 Preliminaries and Setup

*Notation.* We denote the set $\{1, 2, \ldots, n\}$ as $[n]$. Vectors and matrices are represented in bold text (e.g., $\boldsymbol{x}, \boldsymbol{A}$), while scalars are denoted in plain text (e.g., $y$). The input and output domains are symbolized as $\mathcal{X}$ and $\mathcal{Y}$ respectively (unless specified otherwise), and $\boldsymbol{x} \in \mathcal{X}, \boldsymbol{y} \in \mathcal{Y}$ denote the input and output. Additionally, let $\mathcal{F}$ be a set of functions from $\mathcal{X}$ to $\mathcal{Y}$. Consider a transition function $f \in \mathcal{F}$ where $\boldsymbol{y} = f(\boldsymbol{x})$. In this section, we explore the properties of learning $f$, assuming that it can be decomposed into simpler functions $(f_\ell)_{\ell=1}^L$, and thus can be expressed as $f = f_L \circ f_{L-1} \circ \cdots \circ f_1$.

### 2.1 In-context Learning

Following the study by Garg et al. [2022], the fundamental problem of vanilla in-context learning (ICL) involves constructing a prompt with input-output pairs in the following manner:

$$\boldsymbol{p}_n(f) = (\boldsymbol{x}_i, \boldsymbol{y}_i)_{i=1}^n \quad \text{where} \quad \boldsymbol{y}_i = f(\boldsymbol{x}_i). \tag{P-ICL}$$

Here the transition function $f \in \mathcal{F} : \mathcal{X} \rightarrow \mathcal{Y}$ remains constant within a single prompt but can vary across prompts, and the subscript $n$ signifies the number of in-context samples contained in the prompt. Considering language translation as an example, $f$ is identified as the target language, and the prompt can be defined as $\boldsymbol{p}(\texttt{Spanish}) = ((apple, manzana), (ball, pelota), \ldots)$ or $\boldsymbol{p}(\texttt{French})=((cat, chat), (flower, fleur), \ldots)$. Let TF denote any auto-regressive model (e.g., Decoder-only Transformer). The aim of in-context learning is to learn a model that can accurately predict the output, given a prompt $\boldsymbol{p}$ and the test input $\boldsymbol{x}_{\text{test}}$, as shown in the following equation:

$$\texttt{TF}(\boldsymbol{p}_n(\tilde{f}), \boldsymbol{x}_{\text{test}}) \approx \tilde{f}(\boldsymbol{x}_{\text{test}}) \tag{2.1}$$

where $\tilde{f} \in \mathcal{F}$ is the test function which may differ from the functions used during training. Previous work [Zhou et al., 2022, Li et al., 2023c] has demonstrated that longer prompts (containing more examples $n$) typically enhance the performance of the model.

### 2.2 Chain-of-thought Prompt and Prediction

As defined in (P-ICL), the prompt in vanilla ICL only contains input-output pairs of the target function. This demands that the model learns the function $f \in \mathcal{F}$ in one go, which becomes more challenging as $\mathcal{F}$ grows more complex, since larger models and increased prompt length ($n$) are needed to make correct predictions (as depicted by the green curves in Figures 5 and 6). Existing studies on chain-of-thought methods (e.g., [Wei et al., 2022c]) observed that prompts containing step-by-step instructions assist the model in decomposing the function and making better predictions. Specifically, consider a function composed of $L$ subfunctions, represented as $f := f_L \circ f_{L-1} \circ \ldots f_1$. Each intermediate output can be viewed as a step, enabling us to define a length-$n$ CoT prompt related to $f$ with $L$ steps (expressed with $\boldsymbol{s}^\ell, \ell \in [L]$) as follows:

$$\boldsymbol{p}_n(f) = (\boldsymbol{x}_i, \boldsymbol{s}_i^1, \cdots \boldsymbol{s}_i^{L-1}, \boldsymbol{s}_i^L)_{i=1}^n \quad \text{where} \quad \boldsymbol{s}_i^\ell = f_\ell(\boldsymbol{s}_i^{\ell-1}), \ \ell \in [L]. \tag{P-CoT}$$

Here $\boldsymbol{x}_i = \boldsymbol{s}_i^0$, $\boldsymbol{y}_i = \boldsymbol{s}_i^L$ and $f_\ell \in \mathcal{F}_\ell$, which implies that $f \in \mathcal{F}_L \times \cdots \mathcal{F}_1 := \mathcal{F}$.

Next we introduce two methodologies for making predictions within the CoT framework:

**CoT over input only (CoT-I).** Contrasted with ICL, CoT-I considers step-by-step instructions as inputs, nonetheless, the prediction for the last token is performed as a single entity. Our experiments indicate that this approach lowers the sample complexity for TF to comprehend the function $\tilde{f}$ being learned (see the orange curves in Figures 5 and 6). The CoT-I prediction aligns with Eq. (2.1) as follows, while the prompt is determined by (P-CoT).

$$\text{One-shot prediction: } \texttt{TF}(\boldsymbol{p}_n(\tilde{f}), \boldsymbol{x}_{\text{test}}) \approx \tilde{f}(\boldsymbol{x}_{\text{test}}). \tag{2.2}$$

**CoT over both input and output (CoT-I/O).** Despite the fact that CoT-I improves the sample complexity of learning $\tilde{f}$, the TF must still possess the capacity to approximate functions from the function class $\mathcal{F}$, given that the prediction is made in one shot. To mitigate this challenge, we consider

a scenario where in addition to implementing a CoT prompt, we also carry out CoT predictions. Specifically, for a composed problem with inputs formed via (P-CoT), the model recurrently makes $L$-step predictions as outlined below:

$$\text{Step 1: } \text{TF}(\boldsymbol{p}_n(\tilde{f}), \boldsymbol{x}_{\text{test}}) := \hat{\boldsymbol{s}}^1$$

$$\text{Step 2: } \text{TF}(\boldsymbol{p}_n(\tilde{f}), \boldsymbol{x}_{\text{test}}, \hat{\boldsymbol{s}}^1) := \hat{\boldsymbol{s}}^2$$

$$\vdots$$

$$\text{Setp } L: \text{TF}(\boldsymbol{p}_n(\tilde{f}), \boldsymbol{x}_{\text{test}}, \hat{\boldsymbol{s}}^1 \cdots, \hat{\boldsymbol{s}}^{L-1}) \approx \tilde{f}(\boldsymbol{x}_{\text{test}}), \tag{2.3}$$

where at each step (step $\ell$), the model outputs an intermediate step ($\hat{\boldsymbol{s}}^\ell$) which is then fed back to the input sequence to facilitate the next-step prediction ($\hat{\boldsymbol{s}}^{\ell+1}$). Following this strategy, the model only needs to learn the union of the sub-function sets, $\bigcup_{\ell=1}^{L} \mathcal{F}_\ell$, whose complexity scales linearly with the number of steps $L$. Empirical evidence of the benefits of CoT-I/O over ICL and CoT-I in enhancing sample efficiency and model expressivity is reflected in the blue curves shown in Figures 5 and 6.

## 2.3 Model Training

In Figure 1 and Section 2, we have discussed vanilla ICL, CoT-I and CoT-I/O methods. Intuitively, ICL can be viewed as a special case of CoT-I (or CoT-I/O) if we assume only one step is performed. Consequently, we will focus on implementing CoT-I and CoT-I/O for model training in the following.

Consider the CoT prompt as in (P-CoT), and assume that $\boldsymbol{x} \sim \mathcal{D}_{\mathcal{X}}$, and $f_\ell \sim \mathcal{D}_\ell, \ell \in [L]$, where $L$ denotes the number of compositions/steps, such that the final prediction should approximate $f(\boldsymbol{x}) = f_L(f_{L-1} \ldots f_1(\boldsymbol{x})) := \boldsymbol{y} \in \mathcal{Y}$. We define $\ell(\hat{\boldsymbol{y}}, \boldsymbol{y}) : \mathcal{Y} \times \mathcal{Y} \to \mathbb{R}$ as a loss function. For simplicity, we assume $f_\ell(\ldots f_1(\boldsymbol{x})) \in \mathcal{Y}, \ell \in [L]$. Let $N$ represent the in-context window of TF, which implies that TF can only admit a prompt containing up to $N$ in-context samples. Generally, our goal is to ensure high prediction performance given any length-$n$ prompt, where $n \in [N]$. To this end, we train the model using prompts with length from $1$ to $N$ equally and aim to minimize the averaged risk over different prompt size. Assuming the model TF is parameterized by $\boldsymbol{\theta}$ and considering meta learning problem, the objective functions for CoT-I and CoT-I/O are defined as follows.

$$\hat{\boldsymbol{\theta}}^{\text{CoT-I}} = \arg\min_{\boldsymbol{\theta}} \mathbb{E}_{(\boldsymbol{x}_n)_{n=1}^N, (f_\ell)_{\ell=1}^L} \left[ \frac{1}{N} \sum_{n=1}^{N} \ell(\hat{\boldsymbol{y}}_n, f(\boldsymbol{x}_n)) \right]$$

and

$$\hat{\boldsymbol{\theta}}^{\text{CoT-I/O}} = \arg\min_{\boldsymbol{\theta}} \mathbb{E}_{(\boldsymbol{x}_n)_{n=1}^N, (f_\ell)_{\ell=1}^L} \left[ \frac{1}{NL} \sum_{n=1}^{N} \sum_{\ell=1}^{L} \ell(\hat{\boldsymbol{s}}_n^\ell, \boldsymbol{s}_n^\ell) \right]$$

where $\hat{\boldsymbol{y}}_n = \text{TF}(\boldsymbol{p}_n(f), \boldsymbol{x}_n)$ and $\hat{\boldsymbol{s}}_n^\ell = \text{TF}(\boldsymbol{p}_n(f), \boldsymbol{x}_n \cdots \boldsymbol{s}_n^{\ell-1})$. $\boldsymbol{p}_n(f)$ is given by (P-CoT), and as mentioned previously, $\boldsymbol{s}^0 = \boldsymbol{x}$ and $\boldsymbol{s}^L = \boldsymbol{y}$. All $\boldsymbol{x}$ and $f_\ell$ are independent, and we take the expectation of the risk over their respective distributions.

## 3 Provable Approximation of MLPs via Chain-of-Thought

In this section, we present our theoretical findings that demonstrate how CoT-I/O can execute filtering over the CoT prompt, thereby learning a 2-layer MLP with input dimension of $d$ and hidden dimension of $k$, akin to resolving $k$ $d$-dimensional ReLU problems and $1$ $k$-dimensional linear regression problem. Subsequently, in Section 4.1, we examine the performance of CoT-I/O when learning 2-layer random MLPs. Our experiments indicate that CoT-I/O needs only $O(\max(d, k))$ in-context samples to learn the corresponding MLP.

We state our main contribution of establishing a result that decouples CoT-based in-context learning (CoT-I/O) into two phases: (1) *Filtering Phase:* Given a prompt that contains features of multiple MLP layers, retrieve only the features related to a target layer to create an ICL prompt. (2) *ICL Phase:* Given filtered prompt, learn the target layer weights through gradient descent. Combining these two

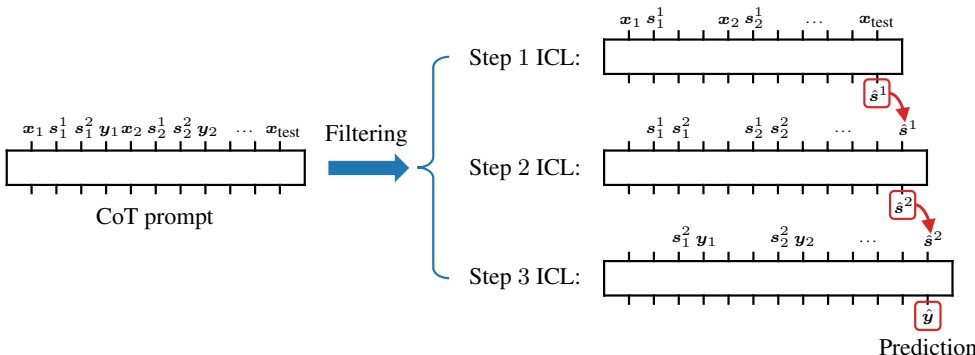

Figure 2: Depiction of Theorem 1 that formalizes CoT as a Filtering + ICL procedure. In this illustration, we utilize in-context samples as depicted in Figure 1, characterizing the CoT prompt by $(\boldsymbol{x}, \boldsymbol{s}^1, \boldsymbol{s}^2, \boldsymbol{y})$. We show a transformer TF, composed of $\mathtt{TF_{LR}} \circ \mathtt{TF_{BE}}$. $\mathtt{TF_{BE}}$ facilitates the implementation of filtering on the CoT prompt (refer to (P-CoT), illustrated on the left side of the figure), subsequently generating vanilla ICL prompts (refer to (P-ICL), illustrated on the right side of the figure). During each step of the ICL process, $\mathtt{TF_{LR}}$ employs gradient descent techniques to address and solve sub-problems, exemplified by linear regression in the context of solving MLPs.

phases, and looping over all layers, we will show that there exists a transformer architecture such that CoT-I/O can provably approximate a multilayer MLP up to a given resolution. An illustration is provided in Figure 2. To state our result, we assume access to an oracle that performs linear regression and consider the consider the condition number of the data matrix.

**Definition 1 (MLP and condition number)** *Consider a multilayer MLP defined by the recursion* $\boldsymbol{s}_i^\ell = \phi(\boldsymbol{W}_\ell \boldsymbol{s}_i^{\ell-1})$ *for* $\ell \in [L]$, $i \in [n]$ *and* $\boldsymbol{s}_i^0 = \boldsymbol{x}_i$. *Here* $\phi(x) := \max(\alpha x, x)$ *is a Leaky-ReLU activation with* $1 \geq \alpha > 0$. *Define the feature matrix* $\boldsymbol{T}_\ell = [\boldsymbol{s}_1^\ell \ \dots \ \boldsymbol{s}_n^\ell]^\top$ *and define its condition number* $\kappa_\ell = \sigma_{\max}(\boldsymbol{T}_\ell)/\sigma_{\min}(\boldsymbol{T}_\ell)$ *(with* $\sigma_{\min} := 0$ *for fat matrices) and* $\kappa_{\max} = \max_{0 \leq \ell < L} \kappa_\ell$.

**Assumption 1 (Oracle Model)** *We assume access to a transformer* $\mathtt{TF_{LR}}$ *which can run* $T$ *steps of gradient descent on the quadratic loss* $\mathcal{L}(\boldsymbol{w}) = \sum_{i=1}^n (y_i - \boldsymbol{w}^\top \boldsymbol{x}_i)^2$ *given a prompt of the form* $(\boldsymbol{x}_1, y_1, \dots, \boldsymbol{x}_n, y_n)$.

We remark that this assumption is realistic and has been formally established by earlier work [Giannou et al., 2023, Akyürek et al., 2022]. Our CoT abstraction builds on these to demonstrate that CoT-I/O can call a blackbox TF model to implement a compositional function when combined with filtering.

We now present our main theoretical contribution. Our result provides a transformer construction that first filters a particular MLP layer from the prompt through the attention mechanism, then applies in-context learning, and repeats this procedure to approximate the MLP output. The precise statement is deferred to the supplementary material.

**Theorem 1 (CoT⇔Filtering+ICL)** *Consider a CoT prompt* $\boldsymbol{p}_n(f)$ *generated from an* $L$-*layer MLP* $f(\cdot)$ *as described in Definition 1, and assume given test example* $(\boldsymbol{x}_{test}, \boldsymbol{s}_{test}^1, \dots \boldsymbol{s}_{test}^L)$. *For any resolution* $\epsilon > 0$, *there exists* $\delta = \delta(\epsilon)$, *iteration choice* $T = \mathcal{O}(\kappa_{\max}^2 \log(1/\epsilon))$, *and a backend transformer construction* $\mathtt{TF_{BE}}$ *such that the concatenated transformer* $\mathtt{TF} = \mathtt{TF_{LR}} \circ \mathtt{TF_{BE}}$ *implements the following: Let* $(\hat{\boldsymbol{s}}^i)_{i=0}^{\ell-1}$ *denote the first* $\ell - 1$ *CoT-I/O outputs of TF where* $\hat{\boldsymbol{s}}^0 = \boldsymbol{x}_{test}$ *and set* $\boldsymbol{p}[\ell] = (\boldsymbol{p}_n(f), \boldsymbol{x}_{test}, \hat{\boldsymbol{s}}^1 \dots \hat{\boldsymbol{s}}^{\ell-1})$. *At step* $\ell$, *TF implements*

1. **Filtering.** *Define the filtered prompt with input/output features of layer* $\ell$,
$$\boldsymbol{p}_n^{filter} = \begin{pmatrix} \dots \boldsymbol{0}, \ \boldsymbol{s}_1^{\ell-1}, \ \boldsymbol{0} \dots \boldsymbol{0}, \ \boldsymbol{s}_n^{\ell-1}, \ \boldsymbol{0} \dots \boldsymbol{0}, \ \hat{\boldsymbol{s}}^{\ell-1} \\ \dots \boldsymbol{0}, \ \ \boldsymbol{0}, \ \ \boldsymbol{s}_1^\ell \dots \boldsymbol{0}, \ \ \ \boldsymbol{0}, \ \ \boldsymbol{s}_n^\ell \dots \boldsymbol{0}, \ \ \boldsymbol{0} \end{pmatrix}.$$
   *There exists a fixed projection matrix* $\boldsymbol{\Pi}$ *that applies individually on tokens such that the backend output obeys* $\|\boldsymbol{\Pi}(\mathtt{TF_{BE}}(\boldsymbol{p}[\ell])) - \boldsymbol{p}_n^{filter}\| \leq \delta$.

2. **Gradient descent.** *The combined model obeys* $\|\mathtt{TF}(\boldsymbol{p}[\ell]) - \boldsymbol{s}_{test}^\ell\| \leq \ell \cdot \epsilon/L$.

$\mathtt{TF_{BE}}$ *has constant number of layers independent of* $n$ *and* $T$. *Consequently, after* $L$ *rounds of CoT-I/O, TF outputs* $f(\boldsymbol{x}_{test})$ *up to* $\epsilon$ *accuracy.*

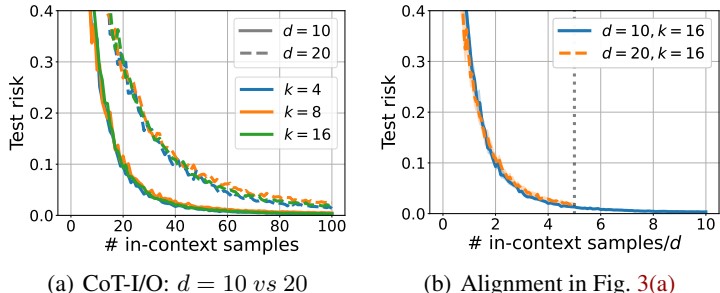

(a) CoT-I/O: $d = 10$ vs $20$        (b) Alignment in Fig. 3(a)

Figure 3: Solving 2-layer MLPs with varying input dimension $d$ and hidden neuron size $k$.

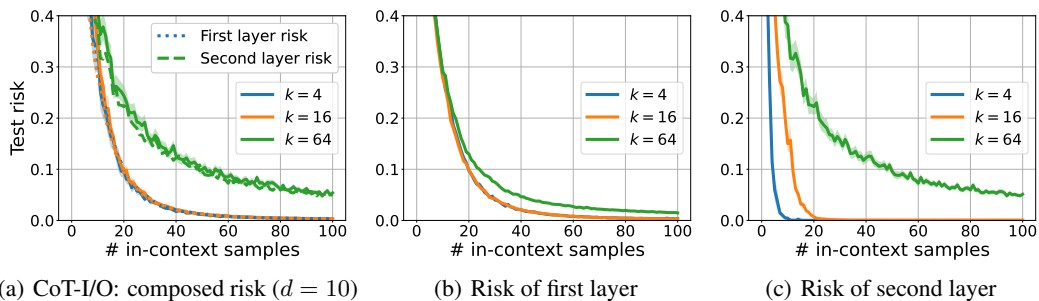

(a) CoT-I/O: composed risk ($d = 10$)     (b) Risk of first layer     (c) Risk of second layer

Figure 4: We decouple the composed risk of predicting 2-layer MLPs into risks of individual layers.

**Remark 1** *Note that, this result effectively shows that, with a sufficiently good blackbox transformer $TF_{LR}$ (per Assumption 1), CoT-I/O can learn an $L$-layer MLP using in-context sample size $n > \max_{\ell \in [L]} d_\ell$ where $d_\ell$ is the input dimension of $\ell$th layer. This is assuming condition number $\kappa_{\max}$ of the problem is finite as soon as all layers become over-determined. Consequently, CoT-I/O needs $\max(k, d)$ sample complexity to learn a two layer MLP. This provides a formal justification for the observation that empirical CoT-I/O performance is agnostic to $k$ as long as $k \leq d$.*

We provide the concrete filtering statements based on the transformer architecture in Appendix A, and the key components of our construction are the following: (i) Inputs are projected through the embedding layer in which a set of encodings, an enumeration of the tokens $(1, 2, \ldots, N)$, an enumeration of the layers $(1, 2, \ldots, L)$ and an identifier for each layer already predicted are all attached. Notice that this "modification" to the input only depends on the sequence length and is agnostic to the token to-be-predicted. This allows for an automated looping over $L$ predictions. (ii) We use this information to extract the sequence length $N$ and the current layer $\ell$ to-be-predicted. (iii) With these at hand, we construct an 'if-then' type of function using the ReLU layers to filter out the samples that are not needed for the prediction.

## 4 Experiments with 2-layer Random MLPs

For a clear exposition, we initially focus on two-layer MLPs, which represent 2-step tasks (e.g., $L = 2$). We begin by validating Theorem 1 using the CoT-I/O method, demonstrating that in-context learning for a 2-layer MLP with $d$ input dimensions and $k$ hidden neurons requires $O(\max(d, k))$ samples. The results are presented in Section 4.1. Subsequently, in Section 4.2, we compare three different methods: ICL, CoT-I, and CoT-I/O. The empirical evidence highlights the advantages of CoT-I/O, showcasing its ability to reduce sample complexity and enhance model expressivity.

**Dataset.** Consider 2-layer MLPs with input $\boldsymbol{x} \in \mathbb{R}^d$, hidden feature (step-1 output) $\boldsymbol{s} \in \mathbb{R}^k$, and output $y \in \mathbb{R}$. Here, $\boldsymbol{s} = f_1(\boldsymbol{x}) := (\boldsymbol{W}\boldsymbol{x})_+$ and $y = f_2(\boldsymbol{s}) := \boldsymbol{v}^\top \boldsymbol{s}$, with $\boldsymbol{W} \in \mathbb{R}^{k \times d}$ and $\boldsymbol{v} \in \mathbb{R}^k$ being the parameters of the first and second layer, and $(x)_+ = \max(x, 0)$ being ReLU activation. The function is composed as $y = \boldsymbol{v}^\top (\boldsymbol{W}\boldsymbol{x})_+$. We define the function distributions as follows: each entry of $\boldsymbol{W}$ is sampled via $\boldsymbol{W}_{ij} \sim \mathcal{N}(0, \frac{2}{k})$, and $\boldsymbol{v} \sim \mathcal{N}(0, \boldsymbol{I}_k)$, with inputs being randomly sampled through $\boldsymbol{x} \sim \mathcal{N}(0, \boldsymbol{I}_d)$[1]. We apply the quadratic loss in our experiments. To avoid the implicit bias due to distribution shift, both training and test datasets are generated following the same strategy.

---

[1]Following this strategy for data generation, the expected norms of $\boldsymbol{x}$, $\boldsymbol{s}$ and $y$ are equivalent, and the risk curves displayed in the figures are normalized for comparison.

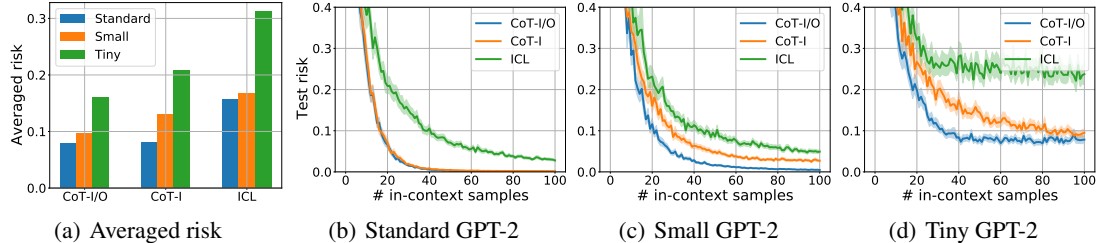

Figure 5: Comparison of the three methods for solving 2-layer MLPs using different GPT-2 models.

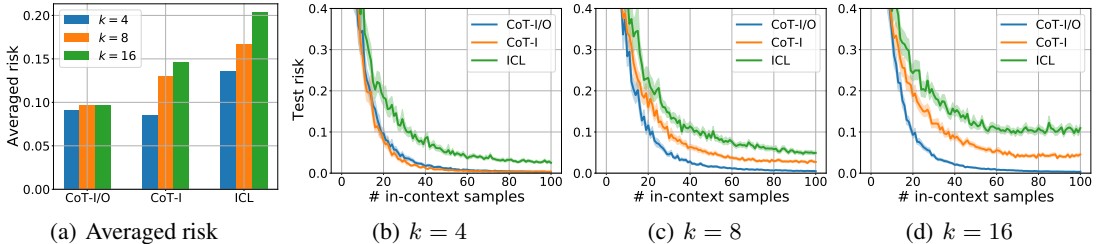

Figure 6: Comparison of the three methods for solving 2-layer MLPs with different hidden sizes.

### 4.1 Empirical Evaluation of CoT-I/O Performance

To investigate how MLP task impacts CoT-I/O performance, we train 2-layer MLPs with varying input dimensions ($d$) and hidden layer sizes ($k$). The results are presented in Figures 3 and 4, and all experiments utilize small GPT-2 models for training[2].

**CoT-I/O performance is agnostic to $k$ when $k \le d$ (Figure 3).** In Fig. 3(a), we train MLPs with $d = 10, 20$ and $k = 4, 8, 16$. Solid and dashed curves represent the CoT-I/O test risk of $d = 10$ and 20 respectively for varying in-context samples. The results indicate that an increase in $d$ will amplifies the number of samples needed for in in-context learning, while the performance remains unaffected by changes in $k \in \{4, 8, 16\}$. To further scrutinize the impact of $d$ on CoT-I/O accuracy, in Fig. 3(b), we adjust the horizontal axis by dividing it by the input dimension $d$, and superimpose both $d = 10, k = 16$ (blue solid) and $d = 20, k = 16$ (orange dashed) results. This alignment of the two curves implies that the in-context sample complexity of CoT-I/O is linearly dependent on $d$.

**Large $k$ dominates CoT-I/O performance (Figure 4).** We further investigate the circumstances under which $k$ begins to govern the CoT-I/O performance. In Figure 4(a), we replicate the same experiments with $d = 10$, but train with wider MLPs ($k = 64$). Blue, orange and green curves represent results for $k = 4, 16, 64$ respectively. Since the hidden dimension $k = 64$ is larger, learning the second layer requires more hidden features ($s$), thus $N = 100$ in-context samples (providing 100 $s$s) are insufficient to fully restore the second layer, leading to performance gaps between $k = 4, 16$ and $k = 64$. To quantify the existing gaps, we conduct single-step evaluations for both the first and the second layers, with the results shown in Figures 4(b) and 4(c). Specifically, let $\boldsymbol{p}_n(\tilde{f})$ be a test prompt containing $n$ in-context samples where $\tilde{f}$ represents any arbitrary 2-layer MLP. Given a test sample $(\boldsymbol{x}_{\text{test}}, \boldsymbol{s}_{\text{test}}, y_{\text{test}})$, the layer predictions are performed as follows.

$$\text{1st layer prediction: } \text{TF}(\boldsymbol{p}_n(\tilde{f}), \boldsymbol{x}_{\text{test}}) := \hat{\boldsymbol{s}},$$

$$\text{2nd layer prediction: } \text{TF}(\boldsymbol{p}_n(\tilde{f}), \boldsymbol{x}_{\text{test}}, \boldsymbol{s}_{\text{test}}) := \hat{y}.$$

The test risks are calculated by $\|\hat{\boldsymbol{s}} - \boldsymbol{s}_{\text{test}}\|^2$ and $(\hat{y} - y_{\text{test}})^2$. The risks illustrated in the figures are normalized for comparability (refer to the appendix for more details). Evidence from Fig. 4(b) and 4(c) shows that while increasing $k$ does not affect the first layer's prediction, it does augment the number of samples required to learn the second layer. Moreover, by plotting the first layer risks of $k = 4, 16$ (blue/orange dotted) and second layer risk of $k = 64$ (green dashed) in Fig. 4(a), we can see that they align with the CoT-I/O composed risks. This substantiates the hypothesis that CoT-I/O learns 2-layer MLP through compositional learning of separate layers.

---

[2]Our code is available at `https://github.com/yingcong-li/Dissecting-CoT`.

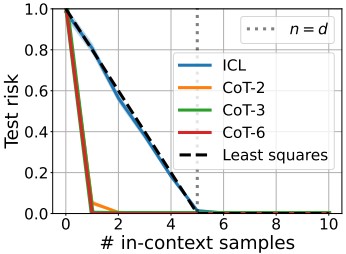
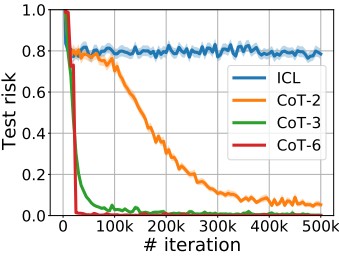

|  (a) Point-to-point meta prediction | (b) One-shot prediction over time |

Figure 7: Evaluations over deep linear MLPs using CoT-I/O and ICL where CoT-$X$ represents the $X$-step CoT-I/O. Fig. 7(a) illustrates point-to-point meta results where the model is trained with substantial number of samples. In contrast, Fig. 7(b) displays the one-shot performance (with only one in-context sample provided) when making predictions during training.

## 4.2 Comparative Analysis of ICL, CoT-I and CoT-I/O

**Varying model sizes (Figure 5).** We initially assess the benefits of CoT-I/O over ICL and CoT-I across different TF models. With $d = 10$ and $k = 8$ fixed, we train three different GPT-2 models: standard, small and tiny GPT-2. The small GPT-2 has 6 layers, 4 attention heads per layer and 128 dimensional embeddings. The standard GPT-2 consists of twice the number of layers, attention heads and embedding dimensionality compared to the small GPT-2, and tiny GPT-2, on the other hand, possesses only half of these hyperparameters compared to the small GPT-2. We evaluate the performance using prompts containing $n$ in-context samples, where $n$ ranges from 1 to $N$ ($N = 100$). The associated test risks are displayed in Figs. 5(b), 5(c) and 5(d). The blue, orange and green curves correspond to CoT-I/O, CoT-I and ICL, respectively. In Fig. 5(a), we present the averaged risks. The results show that using CoT-I/O, the small GPT-2 can solve 2-layer MLPs with approximately 60 samples, while CoT-I requires the standard GPT-2. Conversely, ICL is unable to achieve zero test risk even with the standard GPT-2 model and up to 100 samples. This indicates that to learn 2-layer MLPs in a single shot, ICL requires at least $\mathcal{O}(dk + d)$ samples to restore all function parameters. Conversely, CoT-I and CoT-I/O can leverage implicit samples contained in the CoT prompt. Let $f_1 \in \mathcal{F}_1$ (first layer) and $f_2 \in \mathcal{F}_2$ (second layer). By comparing the performance of CoT-I and CoT-I/O, it becomes evident that the standard GPT-2 is capable of learning the composed function $f = f_2 \circ f_1 \in \mathcal{F}$, which the small GPT-2 cannot express.

**Varying MLP widths (Figure 6).** Next, we explore how different MLP widths impact the performance (by varying the hidden neuron size $k \in \{4, 8, 16\}$). The corresponding results are depicted in Figure 6. The blue, orange and green curves in Fig. 6(b), 6(c) and 6(d) correspond to hidden layer sizes of $k = 4$, 8, and 16, respectively. Fig. 6(a) displays the averaged risks. We keep $d = 10$, $N = 100$ fixed and train with the small GPT-2 model. As discussed in Section 4.1, CoT-I/O can learn a 2-layer MLP using around 60 samples for all $k = 4, 8, 16$ due to its capability to deconstruct composed functions. However, CoT-I can only learn the narrow MLPs with $k = 4$, and ICL is unable to learn any of them. Moreover, we observe a substantial difference in the performance of ICL and CoT-I with varying $k$ (e.g., see averaged risks in Fig. 6(a)). This can be explained by the fact that enlarging $k$ results in more complex $\mathcal{F}_1$ and $\mathcal{F}_2$, thus making the learning of $\mathcal{F} = \mathcal{F}_2 \times \mathcal{F}_1$ more challenging for ICL and CoT-I.

## 5 Further Investigation on Deep Linear MLPs

In Section 4, we have discussed the approximation benefits of CoT-I/O and how it in-context learns 2-layer random MLPs by parallel learning of $k$ $d$-dimensional ReLU and 1 $k$-dimensional linear regression. In this section, we investigate the capability of CoT-I/O in learning longer compositions. For brevity, we will use CoT to refer to CoT-I/O in the rest of the discussion.

**Dataset.** Consider $L$-layer linear MLPs with input $\boldsymbol{x} \in \mathbb{R}^d \sim \mathcal{N}(0, \boldsymbol{I}_d)$, and output generated by $\boldsymbol{y} = \boldsymbol{W}_L \boldsymbol{W}_{L-1} \cdots \boldsymbol{W}_1 \boldsymbol{x}$, where the $\ell$th layer is parameterized by $\boldsymbol{W}_\ell \in \mathbb{R}^{d \times d}$, $\ell \in [L]$. In this work, to better understand the emerging ability of CoT, we assume that each layer draws from the same discrete sub-function set $\bar{\mathcal{F}} = \{\bar{\boldsymbol{W}}_k : \bar{\boldsymbol{W}}_k^\top \bar{\boldsymbol{W}}_k = \boldsymbol{I}, k \in [K]\}$[3]. Therefore, to learn the $L$-layer neural net, CoT only needs to learn $\bar{\mathcal{F}}$ with $|\bar{\mathcal{F}}| = K$, whereas ICL needs to learn the function set $\bar{\mathcal{F}}^L$, which contains $K^L$ random matrices.

---

[3]This assumption ensures that the norm of the feature remains constant across layers ($\|\boldsymbol{x}\| = \|\boldsymbol{y}\| = \|\boldsymbol{s}^\ell\|$), enabling fair evaluation across different layers.

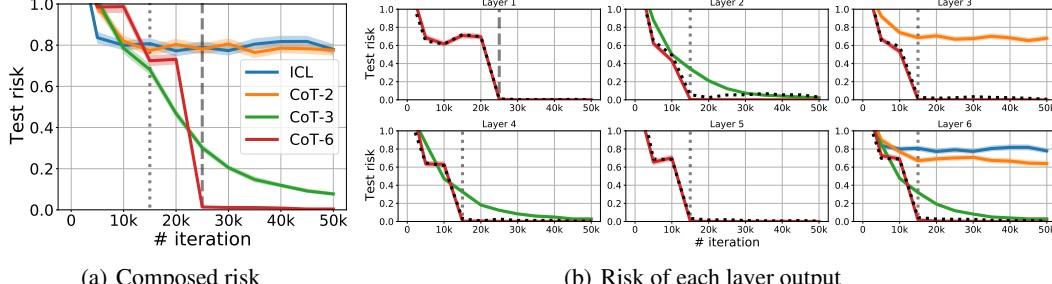

| (a) Composed risk | (b) Risk of each layer output |

Figure 8: Fig. 8(a) is generated by magnifying the initial 50k iterations of Fig. 7(b), and we decouple the composed risks from predicting 6-layer linear MLPs into predictions for each layer, and the results are depicted in Fig. 8(b). Additional implementation details can be found in Section 5.

**Composition ability of CoT (Figures 7).** Set $d = 5$, $L = 6$ and $K = 4$. At each round, we randomly select $L$ matrices $\boldsymbol{W}_\ell$, $\ell \in [L]$ from $\bar{\mathcal{F}}$ so that for any input $\boldsymbol{x}$, we can form a chain

$$\boldsymbol{x} \to \boldsymbol{s}^1 \to \boldsymbol{s}^2 \cdots \to \boldsymbol{s}^6 := \boldsymbol{y},$$

where $\boldsymbol{s}^\ell = \boldsymbol{W}_\ell \boldsymbol{s}^{\ell-1}$, $\ell \in [L]$ and $\boldsymbol{s}^0 := \boldsymbol{x}$. Let CoT-$X$ denote $X$-step CoT-I/O method. For example, the in-context sample of CoT-6 has form of $(\boldsymbol{x}, \boldsymbol{s}^1, \boldsymbol{s}^2, \ldots \boldsymbol{s}^5, \boldsymbol{y})$, which contains all the intermediate outputs from each layer; while CoT-3, CoT-2 have prompt samples formed as $(\boldsymbol{x}, \boldsymbol{s}^2, \boldsymbol{s}^4, \boldsymbol{y})$ and $(\boldsymbol{x}, \boldsymbol{s}^3, \boldsymbol{y})$ respectively. In this setting, ICL is also termed as CoT-1, as its prompt contains only $(\boldsymbol{x}, \boldsymbol{y})$ pairs. To solve the length-6 chain, CoT-$X$ needs to learn a model that can remember $4^{6/X}$ matrices. Therefore, ICL is face a significantly challenge sincd it needs to remember $4^6 = 4,096$ matrices (all combinations of the 4 matrices used for training and testing) compared to just 4 for CoT-6.

We train small GPT-2 models using the CoT-2/-3/-6 and ICL methods, and present the results in Fig. 7(a). As evident from the figure, the performance curves of CoT-2 (orange), CoT-3 (green) and CoT-6 (red) overlap, and they can all make precise predictions in one shot (given an in-context example $n = 1$). It seems that TF has effectively learned to remember up to $64$ matrices (for CoT-2) and compose up to 6 layers (for CoT-6). However, ICL (blue) struggles to learn the 6-layer MLPs in one shot. The black dashed curve shows the solution for linear regression $y = \boldsymbol{\beta}^\top \boldsymbol{x}$ computed directly via least squares given $n$ random training samples, where $\boldsymbol{x}$ is the input and $y$ is from the output of the 6-layer MLPs (e.g., $\boldsymbol{y}[0]$). The test risks for $n = 1, \ldots 10$ are plotted (in Fig. 7(a)), which show that the ICL curve aligns with the least squares performance. This implies that, instead of remembering all $4,096$ matrices, ICL solves the problem from the linear regression phase.

In addition to the meta-learning results which highlight the approximation benefits of multi-step CoT, we also investigate the convergence rate of CoT-2/-3/-6 and ICL, with results displayed in Fig. 7(b). We test the one-shot performance during training and find that CoT-6 converges fastest. This is because it has the smallest sub-function set, and given the same tasks (e.g., deep neural nets), shortening the chain leads to slower convergence. This supports the evidence that taking more steps facilitates faster and more effective learning of complex problems.

**Evidence of Filtering (Figure 8).** As per Theorem 1 and the appendix, transformers can perform filtering over CoT prompts, and the results from 2-layer MLPs align with our theoretical findings. However, can we explicitly observe filtering behaviors? In Fig. 8(a), we display the results of the first 50k iterations from Fig. 7(b), and observe risk drops in CoT-6 (red) at the 15k and 25k iteration (shown as grey dotted and dashed lines). Subsequently, in Fig. 8(b), we plot the test risk of each layer prediction (by feeding the model with correct intermediate features not the predicted ones), where CoT-6 (red) predicts the outputs from all 6 layers ($\boldsymbol{s}^1, \cdots, \boldsymbol{s}^L$). From these figures, we can identify risk drops when predicting different layers, which appear at either 15k (for layer 2, 3, 4, 5, 6) or 25k (for layer 1) iteration. This implies that the model learns to predict each step/layer function independently. Further, we test the filtering evidence of the $\ell$th layer by filling irrelevant positions with random features. Specifically, an in-context example is formed by

$$(\boldsymbol{z}^0, \cdots, \boldsymbol{s}^{\ell-1}, \boldsymbol{s}^\ell, \boldsymbol{z}^{\ell+1}, \ldots \boldsymbol{z}^L), \quad \text{where} \quad \boldsymbol{s}^\ell = \boldsymbol{W}_\ell(\boldsymbol{s}^{\ell-1}) \ \text{and} \ \boldsymbol{z} \sim \mathcal{N}(0, \boldsymbol{I}_d).$$

The test risks are represented by black dotted curves in Fig. 8(b), which aligned precisely with the CoT-6 curves (red). This signifies that each layer's prediction concentrate solely on the corresponding intermediate steps in the prefix, while disregarding irrelevant features. This observation provides evidence that the process of filtering is indeed performed.

# 6 Related Work

With the success of LLMs and prompt structure [Lester et al., 2021], there is growing interest in in-context learning (ICL) from both theoretical and experimental lens [Garg et al., 2022, Brown et al., 2020, von Oswald et al., 2022, Dai et al., 2022, Min et al., 2022, Lyu et al., 2022, Li et al., 2023c, Balim et al., 2023, Xie et al., 2021, Min et al., 2021, Wei et al., 2023, Li et al., 2023a]. As an extension, chain-of-thought (CoT) prompts have made impressive improvements in performing complex reasoning by decomposing it into step-by-step intermediate solutions [Wei et al., 2022c, Narang et al., 2020, Lampinen et al., 2022, Wei et al., 2022b, Zhou et al., 2022, Nye et al., 2021, Veličković and Blundell, 2021, Lanchantin et al., 2023], which in general, shows the ability of transformer in solving compositional functions. Lee et al. [2023], Dziri et al. [2023] study the problem of teaching arithmetic to small transformers and show that breaking the task down into small reasoning steps allows the model to learn faster. Li et al. [2023b] show that small student transformer models can learn from rationalizations sampled from significantly larger teacher models. The idea of learning how to compose skills has been well studied in other literatures [Sahni et al., 2017, Liška et al., 2018]. More specifically, for the problem of learning shallow networks, there are several well known hardness results Goel et al. [2017, 2020], Zhang et al. [2019]. In particular, Hahn and Goyal [2023] shows a formal learnability bound which implies that compositional structure can benefit ICL. However, most of the work focuses on investigating empirical benefits and algorithmic designing of CoT, and there exists little effort studying the underlying mechanisms of CoT.

Considering the expressivity of the transformer architecture itself, Yun et al. [2019] showed that TFs are universal sequence to sequence approximators. More recently, Giannou et al. [2023] use an explicit construction to show that shallow TFs can be used to run general purpose programs as long as we loop them. Other works have also shown the turing-completeness of the TF architecture but these typically require infinite/high precision and recursion around attention layers [Wei et al., 2022a, Pérez et al., 2019, 2021, Liu et al., 2022]. Closer to our work, Akyürek et al. [2022], Von Oswald et al. [2023], von Oswald et al. [2023] prove that a transformer with constant number of layers can implement gradient descent in solving linear regression, and Giannou et al. [2023] introduce similar results by looping outputs back into inputs. Ahn et al. [2023] prove this result from an optimization perspective and show that the global minimum of the training objective implements a single iteration of preconditioned gradient descent for transformers with a single layer of linear attention. Zhou et al. [2023] introduce the RASP-Generalization Conjecture which says that Transformers tend to length generalize on a task if the task can be solved by a short RASP program which works for all input lengths. In this work, we prove CoT can be treated as: first apply filtering on the CoT prompts using special construction, and then in-context learn the filtered prompt.

# 7 Conclusion, Limitations, and Discussion

In this work, we investigate chain-of-thought prompting and shed light on how it enables compositional learning of multilayer perceptrons step-by-step. Specially, we have explored and contrasted three methods: ICL, CoT-I and CoT-I/O, and found that CoT-I/O facilitates better approximation and faster convergence through looping and sample efficiency. Additionally, we empirically and theoretically demonstrated that to learn a 2-layer MLP with $d$-dimensional input and $k$ neurons, CoT-I/O requires $\mathcal{O}(\max(d,k))$ in-context samples whereas ICL runs into approximation error bottlenecks.

While we have provided both experimental and theoretical results to validate the advantages of CoT, it is important to note that our analysis in the main text pertains to in-distribution scenarios. In an effort to address this limitation and demonstrate the robustness of CoT, we have conducted additional simulations, as detailed in Appendix C.1, where the test samples follow a different distribution than the training examples. Also, we note that our focus has been primarily on MLP-based tasks, where the subproblems are essentially instances of simple linear regression. It would be valuable to explore how CoT might influence the training of tasks characterized by more complex structures, longer compositional chains, and a broader variety of subproblems.

There are several interesting avenues for future research to build on our findings. To what extent does our decoupling of CoT (filtering followed by ICL) align with the empirical evidence in practical problems such as code generation and mathematical reasoning? We have shown that CoT-I/O can rely on a linear regression oracle to learn an MLP. To what extent can transformers approximate MLPs without CoT-I/O (e.g. with CoT-I) and what are the corresponding lower/upper bounds?

# 8 Acknowledgements

This work was supported by the NSF CAREER awards #2046816 and #1844951, NSF grant #2212426, AFOSR & AFRL Center of Excellence Award FA9550-18-1-0166, a Google Research Scholar award, an Adobe Data Science Research award, an Army Research Office grant W911NF2110312, and an ONR Grant No. N00014-21-1-2806.

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

# Appendix

## Table of Contents

## A  Construction

### A.1  The Transformer Architecture

For the purpose of this proof, we consider encoder-based transformer architectures and assume that the positional encodings are appended to the input[4]. We also consider that the heads are added and each one has each own key, query and value weight matrices. Formally, we have

$$attn(\boldsymbol{X}) = \boldsymbol{X} + \sum_{h=1}^{H} \boldsymbol{W}_{\mathbf{V}}^{h} \boldsymbol{X}\, softmax((\boldsymbol{W}_{\mathbf{K}}^{h}\boldsymbol{X})^{\top} \boldsymbol{W}_{\mathbf{Q}}^{h}\boldsymbol{X}) \tag{A.1}$$

$$\mathrm{TF}(\boldsymbol{X}) = attn(\boldsymbol{X}) + \boldsymbol{W}_2(\boldsymbol{W}_1 attn(\boldsymbol{X}) + \boldsymbol{b}_1\mathbf{1}_n)_+ + \boldsymbol{b}_2\mathbf{1}_n \tag{A.2}$$

where $\boldsymbol{X} \in \mathbb{R}^{d \times n}$, $H$ is the number of heads used and $f(x) = (x)_+$ is the ReLU activation. We also make use of the temperature $\lambda$, which is a parameter of the softmax. Specifically, $softmax(\boldsymbol{x}) = \{e^{\lambda x_i}/\sum_j e^{\lambda x_j}\}_i$. Notice that as $\lambda \to \infty$ we have $softmax(\boldsymbol{x}) \to \max_i x_i$. We also assume that the inputs are bounded; we denote with $N_{max}$ the maximum sequence length of the model.

**Assumption 2** *Each entry is bounded by some constant $c$, which implies that $\|\boldsymbol{X}\| \leq c'$, for some large $c'$ that depends on the maximum sequence length and the width of the transformer.*

### A.2  Positional Encodings

In the constructions below we use a combination of the binary representation of each position, as well as some additional information (e.g. 0-1 bits) as described in the following sections. The binary representations and in general encodings we construct, require only logarithmic space with respect

---

[4]We note here that in terms of our construction adding the encodings or appending them to the input can be viewed in a similar manner. Specifically, we can consider that the up-projection step projects the input to zero padded vectors, while the encodings are orthogonal to that projection in the sense that the have zero in the non-zero coordinates of the input. In that case adding the positional encodings corresponds to appending them to the input.

to the sequence length. Notice that binary representation of the positions satisfy the following two conditions:

1. Let $\mathbf{r}_i$ be the binary representation of the $i$-th position, then there exists $\varepsilon > 0$ such that $\mathbf{r}_i^\top \mathbf{r}_i > \mathbf{r}_i^\top \mathbf{r}_j + \varepsilon$ for all $j \neq i$.

2. There exists a one layer transformer that can implement the addition of two pointers (see Lemma 2 ).

## A.3 Constructing Some Useful "Black-box" Functions

We follow the construction of previous work [Giannou et al., 2023] and use the following individual implementations as they do. We repeat the statements here for convenience. The first lemma is also similar to Akyürek et al. [2022], we however follow a slightly different proof.

**Lemma 1** *A transformer with one layer, two heads, and embedding dimension of $\mathcal{O}(\log n + d)$, where $d$ is the input dimension and $n$ is the sequence length, can copy any block of the input to any other block of the input with error $\varepsilon$ arbitrarily small.*

**Proof.** Consider an input of the following form

$$
\boldsymbol{X} = \begin{bmatrix} \boldsymbol{x}_1 & \boldsymbol{x}_2 & \dots & \boldsymbol{x}_n \\ \boldsymbol{y}_1 & \boldsymbol{y}_2 & \dots & \boldsymbol{y}_n \\ \mathbf{r}_1 & \mathbf{r}_2 & \dots & \mathbf{r}_n \\ \mathbf{r}_k & \mathbf{r}_2 & \dots & \mathbf{r}_n \end{bmatrix} \tag{A.3}
$$

where $\boldsymbol{x}_i, \boldsymbol{y}_i \in \mathbb{R}^d$ are column vectors for all $i = 1, \dots, n$. We present how we can move a block of data from the $(d+1 : 2d, k)$ position block to the $(1 : d, 1)$ position block, meaning to move the point $\boldsymbol{y}_k$ to the position of the point $\boldsymbol{x}_1$. It is straightforward to generalize the proof that follows to move blocks from $(i : i + k, j)$ to $(i' : i' + k, j')$ for arbitrary $i, i', j, j', k$.

Let in Eq. A.1 $\boldsymbol{W}_{\mathbf{K}}^1 = [\mathbf{0} \quad \mathbf{0} \quad \boldsymbol{I} \quad \mathbf{0}]$ and $\boldsymbol{W}_{\mathbf{Q}}^1 = [\mathbf{0} \quad \mathbf{0} \quad \mathbf{0} \quad \boldsymbol{I}]$ and $\boldsymbol{W}_{\mathbf{K}}^2 = [\mathbf{0} \quad \mathbf{0} \quad \mathbf{0} \quad \boldsymbol{I}]$ and $\boldsymbol{W}_{\mathbf{Q}}^2 = [\mathbf{0} \quad \mathbf{0} \quad \mathbf{0} \quad \boldsymbol{I}]$ then we have

$$
(\boldsymbol{W}_{\mathbf{K}}^1 \boldsymbol{X})^\top (\boldsymbol{W}_{\mathbf{Q}}^1 \boldsymbol{X}) = \begin{bmatrix} \mathbf{r}_1^\top \mathbf{r}_k & \mathbf{r}_1^\top \mathbf{r}_2 & \dots & \mathbf{r}_1^\top \mathbf{r}_k & \dots & \mathbf{r}_1^\top \mathbf{r}_n \\ \mathbf{r}_2^\top \mathbf{r}_k & \mathbf{r}_2^\top \mathbf{r}_2 & \dots & \mathbf{r}_2^\top \mathbf{r}_k & \dots & \mathbf{r}_2^\top \mathbf{r}_n \\ \vdots & \vdots & \ddots & \vdots & \ddots & \vdots \\ \mathbf{r}_k^\top \mathbf{r}_k & \mathbf{r}_k^\top \mathbf{r}_2 & \dots & \mathbf{r}_k^\top \mathbf{r}_k & \dots & \mathbf{r}_k^\top \mathbf{r}_n \\ \vdots & \vdots & \ddots & \vdots & \ddots & \vdots \\ \mathbf{r}_n^\top \mathbf{r}_k & \mathbf{r}_n^\top \mathbf{r}_2 & \dots & \mathbf{r}_n^\top \mathbf{r}_k & \dots & \mathbf{r}_n^\top \mathbf{r}_n \end{bmatrix} \tag{A.4}
$$

As $\lambda \to \infty$ and after the application of the softmax operator the above matrix becomes equal to $[\mathrm{e}_k \quad \mathrm{e}_2 \quad \dots \quad \mathrm{e}_k \quad \dots \mathrm{e}_n] + \epsilon \mathbf{M}$, where $\mathrm{e}_i$ is the one-hot vector with 1 in the $k$-th position, $\|M\| \leq 1$ and $\epsilon$ is controllable by the temperature parameter and can be arbitrary small. Let finally $\boldsymbol{W}_{\mathbf{V}}^1 = \begin{bmatrix} \mathbf{0} & \boldsymbol{I} & \mathbf{0} & \mathbf{0} \\ \mathbf{0} & \mathbf{0} & \mathbf{0} & \mathbf{0} \end{bmatrix}$ we get that

$$
\boldsymbol{W}_{\mathbf{V}}^1 \boldsymbol{X} \, softmax((\boldsymbol{W}_{\mathbf{K}}^1 \boldsymbol{X})^\top (\boldsymbol{W}_{\mathbf{Q}}^1 \boldsymbol{X})) = \begin{bmatrix} \boldsymbol{y}_k & \boldsymbol{x}_2 & \dots & \boldsymbol{x}_n \\ \mathbf{0} & \mathbf{0} & \dots & \mathbf{0} \end{bmatrix} + \epsilon \mathbf{M} \tag{A.5}
$$

By repeating the exact same steps for the second head and letting $\boldsymbol{W}_{\mathbf{V}}^2 = \begin{bmatrix} -\boldsymbol{I} & \mathbf{0} & \mathbf{0} & \mathbf{0} \\ \mathbf{0} & \mathbf{0} & \mathbf{0} & \mathbf{0} \end{bmatrix}$ and adding back the residual we get the desired result. ∎

A slightly different implementation can be found in Giannou et al. [2023], in which the ReLU layers are also used, together with indicator vectors that define whether a position should be updated or not.

**Lemma 2** *There exists a 1-hidden layer feedforward, ReLU network, with $8d$ activations in the hidden layer and $d$ neurons in the output layer that when given two $d$-dimensional binary vectors representing two non-negative integers, can output the binary vector representation of their sum, as long as the sum is less than $2^{d+1}$.*

**Lemma 3** *Let* $\mathbf{A} \in \mathbb{R}^{d \times m}$ *and* $\mathbf{B} \in \mathbb{R}^{d \times n}$. *Then for any* $\epsilon > 0$ *there exists a transformer-based function block with 2 layers, 1 head and width* $r = O(d)$ *that outputs the multiplication* $\mathbf{A}^\top \mathbf{B} + \epsilon \mathbf{M}$, *for some* $\|\mathbf{M}\| \leq 1$.

**Remark 2** *Notice that based on the proof of this lemma, the matrices/scalars/vectors need to be in the same rows, i.e.,* $\mathbf{Q} = [\mathbf{A} \quad \mathbf{B}]$. *By also appending the appropriate binary encodings we can move the output at any specific place we choose as in Lemma 1. Also, following the proof of the paper the input matrix is*

$$X = \begin{bmatrix} \mathbf{Q} & \mathbf{0} & \mathbf{0} \\ \mathbf{0} & \mathbf{11}^\top & \mathbf{0} \\ \mathbf{I} & \mathbf{0} & \mathbf{0} \\ & \mathbf{r}^{(1)} & \\ & \mathbf{r}^{(2)} & \end{bmatrix}$$

*where* $\mathbf{r}^{(1)}, \mathbf{r}^{(2)}$ *are chosen as to specify the position of the result.*

### A.4 Results on Filtering

**Lemma 4** *Assume that the input to a transformer layer is of the following form*

$$X = \begin{bmatrix} \boldsymbol{x}_1 & \boldsymbol{x}_2 & \ldots & \boldsymbol{x}_{n-1} & \boldsymbol{x}_n \\ \mathbf{0} & \mathbf{0} & \ldots & \mathbf{0} & \mathbf{0} \\ b_1 & b_2 & \ldots & b_{n-1} & b_n \\ b_1' & b_2' & \ldots & b_{n-1}' & b_n' \end{bmatrix} \tag{A.6}$$

*where* $b_i, b_i' \in \{0, 1\}$, *with zero indicating that the corresponding point should be ignored. Then there exists a transformer TF consisting only of a ReLU layer that performs this filtering, i.e.,*

$$TF(X) = \begin{bmatrix} \mathbf{1}\{b_1 \neq 0\}\boldsymbol{x}_1 & \mathbf{1}\{b_2 \neq 0\}\boldsymbol{x}_2 & \ldots & \mathbf{1}\{b_{n-1} \neq 0\}\boldsymbol{x}_{n-1} & \mathbf{1}\{b_n \neq 0\}\boldsymbol{x}_n \\ \mathbf{1}\{b_1' \neq 0\}\boldsymbol{x}_1 & \mathbf{1}\{b_2' \neq 0\}\boldsymbol{x}_2 & \ldots & \mathbf{1}\{b_{n-1}' \neq 0\}\boldsymbol{x}_{n-1} & \mathbf{1}\{b_n' \neq 0\}\boldsymbol{x}_n \\ b_1 & b_2 & \ldots & b_{n-1} & b_n \\ b_1' & b_2' & \ldots & b_{n-1}' & b_n' \end{bmatrix} \tag{A.7}$$

**Proof.** The layer is the following:

$$\text{TF}(\boldsymbol{x}_i) = x_i + (-Cb_i - \boldsymbol{x}_i)_+ - (-Cb_i + \boldsymbol{x}_i)_+ \tag{A.8}$$

*for some large constant* $C$. Notice that if $b_i = 1$ the output is just $\boldsymbol{x}_i$. But if $b_i = 0$ then the output is zero. For the second set instead of using the bits $b_i$, we use the $b_i'$. ∎

**Remark 3** *Notice that if some* $b_i$ *is instead of* $1$, $1 \pm \varepsilon$, $\varepsilon < c/C$. *Then the output of the above layer would be*

$$TF(\boldsymbol{x}_i) = \boldsymbol{x}_i + (-C \pm C\varepsilon - \boldsymbol{x}_i) - (-C \pm C\varepsilon + \boldsymbol{x}_i) \tag{A.9}$$
$$= \boldsymbol{x}_i \tag{A.10}$$

*while if some* $b_i = \pm\varepsilon$ *instead of zero, and assuming that* $\boldsymbol{x}_i > c > 0$ *or* $\boldsymbol{x}_i < -c < 0$ *the output would be*

$$TF(\boldsymbol{x}_i) = \boldsymbol{x}_i + (-C\varepsilon - \boldsymbol{x}_i)_+ - (-Cb_i + \boldsymbol{x}_i)_+ \tag{A.11}$$
$$= \boldsymbol{x}_i \pm C\varepsilon - \boldsymbol{x}_i \tag{A.12}$$
$$\leq c \tag{A.13}$$

*If* $|\boldsymbol{x}_i| \leq c$, *again the output would be less than or equal to* $c$, *where* $c$ *can be arbitrarily small.*

Our target is to create these binary tokens, as to perform the filtering. We now describe for clarity how next word prediction is performed. An $d \times N$-dimensional input is given to the transformer architecture, which is up-projected using the embedding layer; the positional encodings are also added/appended to the input. At the last layer, the last token predicted is appended to the initial input to the transformer architecture. The only difference of the new input including the positional encodings ( the input for the next iteration ) is the $n + 1$-th token. This is a property that our construction maintains, *i.e.,* the positional encodings used are oblivious to the prediction step performed and are

always the same for each individual token. We consider encoder-based architectures in all of our lemmas below.

In the subsequent lemma we construct an automated process that works along those guidelines. To do so we assume that the input to the transformer contains the following information:

1. An enumeration of the tokens from $1$ to $N$.

2. The $\ln$ of the above enumeration.

3. Zeros for the tokens that correspond to the data points, $1$ for each token that is the $x_{\text{test}}$ or it is a prediction based on it.

4. An enumeration $1$ to $L$ for each one of the data points provided. For example, if we are given three sets of data we would have : $1 \ldots L\ 1 \ldots L\ 1 \ldots L$.

5. Some extra information that is needed to implement a multiplication step as described in Lemma 3 and to move things to the correct place.

The above information can be viewed as part of the encodings that are appended to the input of the transformer. Formally, we have

**Lemma 5** *Consider that a prompt with $n$ in-context samples is given and the $\ell - 1$-th prediction has been made, and the transformer is to predict the $\ell$-th one. Assume the input to the transformer is:*

$$
X = \left[
\begin{array}{ccccccccccc}
x_1 & \ldots & s_1^{\ell-1} & s_1^{\ell} & \ldots & s_2^{\ell-1} & s_2^{\ell} & \ldots & s_n^{L} & x_{test} & \ldots & \hat{s}^{\ell-1} \\
0 & \ldots & 0 & 0 & \ldots & 0 & 0 & \ldots & 0 & 0 & \ldots & 0 \\
\hline
1 & \ldots & \ell & \ell+1 & \ldots & \ell & \ell+1 & \ldots & L+1 & 1 & \ldots & \ell \\
1 & \ldots & \ell & \ell+1 & \ldots & \ell & \ell+1 & \ldots & L+1 & 1 & \ldots & \ell \\
\hline
1 & \ldots & \ell & \ell+1 & \ldots & L+\ell+1 & L+\ell+2 & \ldots & n(L+1) & n(L+1)+1 & \ldots & N \\
\ln(1) & \ldots & \ln(\ell) & \ln(\ell+1) & \ldots & \ln(L+\ell+1) & \ln(L+\ell+2) & \ldots & \ln(n(L+1)) & \ln(n(L+1)+1) & \ldots & \ln(N) \\
0 & \ldots & 0 & 0 & \ldots & 0 & 0 & \ldots & 0 & 1 & \ldots & 1 \\
1 & \ldots & 1 & 1 & \ldots & 1 & 1 & \ldots & 1 & 1 & \ldots & 1 \\
m_1 & \ldots & m_\ell & m_{\ell+1} & \ldots & m_{L+\ell+1} & m_{L+\ell+2} & \ldots & m_{n(L+1)} & m_{n(L+1)+1} & \ldots & m_N
\end{array}
\right]
$$

*where $(\hat{s}^i)_{i=1}^{\ell-1}$ denote the first $\ell - 1$ recurrent outputs of TF and for simplicity, let $N := n(L+1) + \ell$ denote the total number of tokens. Then there exists a transformer TF consisting of 7 layers that has as output*

$$
TF(X) = \left[
\begin{array}{ccccccccccc}
0 & \ldots & s_1^{\ell-1} & 0 & \ldots & s_2^{\ell-1} & 0 & \ldots & 0 & 0 & \ldots & \hat{s}^{\ell-1} \\
0 & \ldots & 0 & s_1^{\ell} & \ldots & 0 & s_2^{\ell} & \ldots & 0 & 0 & \ldots & 0 \\
\hline
0 & \ldots & 1 & 0 & \ldots & 1 & 0 & \ldots & 0 & 0 & \ldots & 1 \\
0 & \ldots & 0 & 1 & \ldots & 0 & 1 & \ldots & 0 & 0 & \ldots & 0 \\
\hline
1 & \ldots & \ell & \ell+1 & \ldots & L+\ell+1 & L+\ell+2 & \ldots & n(L+1) & n(L+1)+1 & \ldots & N \\
\ln(1) & \ldots & \ln(\ell) & \ln(\ell+1) & \ldots & \ln(L+\ell+1) & \ln(L+\ell+2) & \ldots & \ln(n(L+1)) & \ln(n(L+1)+1) & \ldots & \ln(N) \\
0 & \ldots & 0 & 0 & \ldots & 0 & 0 & \ldots & 0 & 1 & \ldots & 1 \\
1 & \ldots & 1 & 1 & \ldots & 1 & 1 & \ldots & 1 & 1 & \ldots & 1 \\
m_1 & \ldots & m_\ell & m_{\ell+1} & \ldots & m_{L+\ell+1} & m_{L+\ell+2} & \ldots & m_{n(L+1)} & m_{n(L+1)+1} & \ldots & m_N
\end{array}
\right]
$$

*with error up to $\delta \mathbf{M}$, where $\|\mathbf{M}\| \leq 1$ and $\delta > 0$ is a constant that is controlled and can be arbitrarily small.*

**Proof.**

**Step 1: Extract the sequence length (1 layer).** Let

$$
W_{\mathbf{K}} = \begin{bmatrix} 0 & 0 & 0 & 0 & 0 & 1 & 0 & 0 & 0 \end{bmatrix} \quad W_{\mathbf{Q}} = \begin{bmatrix} 0 & 0 & 0 & 0 & 0 & 0 & 0 & 1 & 0 \end{bmatrix} \quad \text{(A.14)}
$$

and thus

$$(\boldsymbol{W_K X})^\top (\boldsymbol{W_Q X}) = \begin{bmatrix} \ln(1) & \ln(1) & \dots & \ln(1) \\ \ln(2) & \ln(2) & \dots & \ln(2) \\ \vdots & \vdots & \ddots & \vdots \\ \ln(\mathrm{N}) & \ln(\mathrm{N}) & \dots & \ln(\mathrm{N}) \end{bmatrix}. \tag{A.15}$$

So after the softmax is applied we have

$$softmax((\boldsymbol{W_K X})^\top (\boldsymbol{W_Q X})) = \begin{bmatrix} \dfrac{1}{\sum_{i=1}^{\mathrm{N}} i} & \dfrac{1}{\sum_{i=1}^{\mathrm{N}} i} & \cdots & \dfrac{1}{\sum_{i=1}^{\mathrm{N}} i} \\ \dfrac{2}{\sum_{i=1}^{\mathrm{N}} i} & \dfrac{2}{\sum_{i=1}^{\mathrm{N}} i} & \cdots & \dfrac{2}{\sum_{i=1}^{\mathrm{N}} i} \\ \vdots & \vdots & \ddots & \vdots \\ \dfrac{\mathrm{N}}{\sum_{i=1}^{\mathrm{N}} i} & \dfrac{\mathrm{N}}{\sum_{i=1}^{\mathrm{N}} i} & \cdots & \dfrac{\mathrm{N}}{\sum_{i=1}^{\mathrm{N}} i} \end{bmatrix}. \tag{A.16}$$

We then set the weight value matrix as to zero-out all lines except for one line as follows

$$\boldsymbol{W_V X} = \begin{bmatrix} \boldsymbol{0} & \boldsymbol{0} & \dots & \boldsymbol{0} \\ 1 & 2 & \dots & \mathrm{N} \\ \boldsymbol{0} & \boldsymbol{0} & \dots & \boldsymbol{0} \end{bmatrix}. \tag{A.17}$$

After adding the residual and using an extra head where the softmax returns identity matrix and the value weight matrix is minus the identity, we get attention output

$$\begin{bmatrix} \boldsymbol{x}_1 & \dots & \boldsymbol{s}_1^{\ell-1} & \boldsymbol{s}_1^{\ell} & \dots & \boldsymbol{s}_2^{\ell-1} & \boldsymbol{s}_2^{\ell} & \dots & \boldsymbol{s}_n^L & \boldsymbol{x}_{\text{test}} & \dots & \hat{\boldsymbol{s}}^{\ell-1} \\ \boldsymbol{0} & \dots & \boldsymbol{0} & \boldsymbol{0} & \dots & \boldsymbol{0} & \boldsymbol{0} & \dots & \boldsymbol{0} & \boldsymbol{0} & \dots & \boldsymbol{0} \\ 1 & \dots & \ell & \ell+1 & \dots & \ell & \ell+1 & \dots & L+1 & 1 & \dots & \ell \\ 1 & \dots & \ell & \ell+1 & \dots & \ell & \ell+1 & \dots & L+1 & 1 & \dots & \ell \\ \frac{\sum_{i=1}^{\mathrm{N}} i^2}{\sum_{i=1}^{\mathrm{N}} i} & \dots & \frac{\sum_{i=1}^{\mathrm{N}} i^2}{\sum_{i=1}^{\mathrm{N}} i} & \frac{\sum_{i=1}^{\mathrm{N}} i^2}{\sum_{i=1}^{\mathrm{N}} i} & \dots & \frac{\sum_{i=1}^{\mathrm{N}} i^2}{\sum_{i=1}^{\mathrm{N}} i} & \frac{\sum_{i=1}^{\mathrm{N}} i^2}{\sum_{i=1}^{\mathrm{N}} i} & \dots & \frac{\sum_{i=1}^{\mathrm{N}} i^2}{\sum_{i=1}^{\mathrm{N}} i} & \frac{\sum_{i=1}^{\mathrm{N}} i^2}{\sum_{i=1}^{\mathrm{N}} i} & \dots & \frac{\sum_{i=1}^{\mathrm{N}} i^2}{\sum_{i=1}^{\mathrm{N}} i} \\ \ln(1) & \dots & \ln(\ell) & \ln(\ell+1) & \dots & \ln(L+\ell+1) & \ln(L+\ell+2) & \dots & \ln(n(L+1)) & \ln(n(L+1)+1) & \dots & \ln(\mathrm{N}) \\ 0 & \dots & 0 & 0 & \dots & 0 & 0 & \dots & 0 & 1 & \dots & 1 \\ 1 & \dots & 1 & 1 & \dots & 1 & 1 & \dots & 1 & 1 & \dots & 1 \\ \boldsymbol{m}_1 & \dots & \boldsymbol{m}_\ell & \boldsymbol{m}_{\ell+1} & \dots & \boldsymbol{m}_{L+\ell+1} & \boldsymbol{m}_{L+\ell+2} & \dots & \boldsymbol{m}_{n(L+1)} & \boldsymbol{m}_{n(L+1)+1} & \dots & \boldsymbol{m}_{\mathrm{N}} \end{bmatrix}.$$

Notice that $\dfrac{\sum_{i=1}^{\mathrm{N}} i^2}{\sum_{i=1}^{\mathrm{N}} i} = \dfrac{2\mathrm{N}(\mathrm{N}+1)(2\mathrm{N}+1)}{6\mathrm{N}(\mathrm{N}+1)} = \dfrac{2\mathrm{N}+1}{3}$. We then use the ReLU layer as to multiply with $3/2$ and subtract 1 from this column. This results to attention output

$$\begin{bmatrix} \boldsymbol{x}_1 & \dots & \boldsymbol{s}_1^{\ell-1} & \boldsymbol{s}_1^{\ell} & \dots & \boldsymbol{s}_2^{\ell-1} & \boldsymbol{s}_2^{\ell} & \dots & \boldsymbol{s}_n^L & \boldsymbol{x}_{\text{test}} & \dots & \hat{\boldsymbol{s}}^{\ell-1} \\ \boldsymbol{0} & \dots & \boldsymbol{0} & \boldsymbol{0} & \dots & \boldsymbol{0} & \boldsymbol{0} & \dots & \boldsymbol{0} & \boldsymbol{0} & \dots & \boldsymbol{0} \\ 1 & \dots & \ell & \ell+1 & \dots & \ell & \ell+1 & \dots & L+1 & 1 & \dots & \ell \\ 1 & \dots & \ell & \ell+1 & \dots & \ell & \ell+1 & \dots & L+1 & 1 & \dots & \ell \\ \mathrm{N} & \dots & \mathrm{N} & \mathrm{N} & \dots & \mathrm{N} & \mathrm{N} & \dots & \mathrm{N} & \mathrm{N} & \dots & \mathrm{N} \\ \ln(1) & \dots & \ln(\ell) & \ln(\ell+1) & \dots & \ln(L+\ell+1) & \ln(L+\ell+2) & \dots & \ln(n(L+1)) & \ln(n(L+1)+1) & \dots & \ln(\mathrm{N}) \\ 0 & \dots & 0 & 0 & \dots & 0 & 0 & \dots & 0 & 1 & \dots & 1 \\ 1 & \dots & 1 & 1 & \dots & 1 & 1 & \dots & 1 & 1 & \dots & 1 \\ \boldsymbol{m}_1 & \dots & \boldsymbol{m}_\ell & \boldsymbol{m}_{\ell+1} & \dots & \boldsymbol{m}_{L+\ell+1} & \boldsymbol{m}_{L+\ell+2} & \dots & \boldsymbol{m}_{n(L+1)} & \boldsymbol{m}_{n(L+1)+1} & \dots & \boldsymbol{m}_{\mathrm{N}} \end{bmatrix}$$

Notice that this step does not involve any error.

**Step 2: Extract the identifier of the prediction to be made $\ell$ (3 layers).** Now, by setting the key and query weight matrices of the attention as to just keep the all ones row, we get a matrix that attends

equally to all tokens. Then the value weight matrix keeps the row that contains $\ell$ ones and thus we get the number $\dfrac{\ell}{N}$, propagated in all the sequence length. Then the attention output is as follows:

$$
\left[\begin{array}{ccccccccccc}
\boldsymbol{x}_1 & \dots & \boldsymbol{s}_1^{\ell-1} & \boldsymbol{s}_1^{\ell} & \dots & \boldsymbol{s}_2^{\ell-1} & \boldsymbol{s}_2^{\ell} & \dots & \boldsymbol{s}_n^{L} & \boldsymbol{x}_{\text{test}} & \dots & \hat{\boldsymbol{s}}^{\ell-1} \\
\mathbf{0} & \dots & \mathbf{0} & \mathbf{0} & \dots & \mathbf{0} & \mathbf{0} & \dots & \mathbf{0} & \mathbf{0} & \dots & \mathbf{0} \\
\hline
1 & \dots & \ell & \ell+1 & \dots & \ell & \ell+1 & \dots & L+1 & 1 & \dots & \ell \\
1 & \dots & \ell & \ell+1 & \dots & \ell & \ell+1 & \dots & L+1 & 1 & \dots & \ell \\
\hline
N & \dots & N & N & \dots & N & N & \dots & N & N & \dots & N \\
\dfrac{\ell}{N} & \dots & \dfrac{\ell}{N} & \dfrac{\ell}{N} & \dots & \dfrac{\ell}{N} & \dfrac{\ell}{N} & \dots & \dfrac{\ell}{N} & \dfrac{\ell}{N} & \dots & \dfrac{\ell}{N} \\
0 & \dots & 0 & 0 & \dots & 0 & 0 & \dots & 0 & 1 & \dots & 1 \\
1 & \dots & 1 & 1 & \dots & 1 & 1 & \dots & 1 & 1 & \dots & 1 \\
\boldsymbol{m}_1 & \dots & \boldsymbol{m}_\ell & \boldsymbol{m}_{\ell+1} & \dots & \boldsymbol{m}_{L+\ell+1} & \boldsymbol{m}_{L+\ell+2} & \dots & \boldsymbol{m}_{n(L+1)} & \boldsymbol{m}_{n(L+1)+1} & \dots & \boldsymbol{m}_N
\end{array}\right]. \quad \text{(A.18)}
$$

So far this step does not involve any error. As described in Lemma 3 to implement multiplication of two values up to any error $\epsilon$, we need 1) have the two numbers to be multiplied next to each other and an extra structure (some constants, which we consider that are encoded in the last rows of the matrix $\boldsymbol{m}_i$) and 2) the corresponding structure presented in Lemma 3 . So we need to make the following transformation in the rows containing N and $\dfrac{\ell}{N}$

$$
\begin{bmatrix}
N & N & \dots & N \\
\dfrac{\ell}{N} & \dfrac{\ell}{N} & \dots & \dfrac{\ell}{N}
\end{bmatrix}
\rightarrow
\begin{bmatrix}
N & \dfrac{\ell}{N} & \dots & N \\
\dfrac{\ell}{N} & \dfrac{\ell}{N} & \dots & \dfrac{\ell}{N}
\end{bmatrix}
\tag{A.19}
$$

$$
\rightarrow
\begin{bmatrix}
\ell & * & \dots & * \\
* & * & \dots & *
\end{bmatrix}
\tag{A.20}
$$

$$
\rightarrow
\begin{bmatrix}
\ell & \ell & \dots & \ell \\
* & * & \dots & *
\end{bmatrix}
\tag{A.21}
$$

where $*$ denotes inconsequential values. For the first step we assume that we have the necessary binary representations [5] and then we use Lemma 1 which shows how we can perform the operation of copy/paste. This step involves an error and the current output would be $\tilde{\boldsymbol{X}}_1 = \boldsymbol{X}^* + \varepsilon_1 \boldsymbol{M}_1$, where $\varepsilon_1$ is controlled and we will determine it in the sequence, $\|\boldsymbol{M}_1\| \leq 1$ and $\boldsymbol{X}_1^*$ is the desired output. For the second step now we use Lemma 3 to perform the multiplication, this will affect some of the other values. To analyze the error of this step, we know that given $\boldsymbol{X}_1^*$ we will get the desired output $\boldsymbol{X}_2^* + \varepsilon_2 \boldsymbol{M}_2$, where $\|\boldsymbol{M}_2\| \leq 1$ and $\varepsilon_2$ is to be determined. However, are given $\tilde{X}_1$ in the multiplication procedure[6] which results in the following output $\tilde{\boldsymbol{X}}_2 = (\tilde{\boldsymbol{X}}_1)^\top \tilde{\boldsymbol{X}}_1 + \varepsilon_2 \boldsymbol{M}_2 = \boldsymbol{X}_1^{*\top} \boldsymbol{X}_1^* + \varepsilon_1 \boldsymbol{M}_1^\top \boldsymbol{X}_1^* + \varepsilon_1 \boldsymbol{M}_1^\top \boldsymbol{M}_1 + \varepsilon_2 \boldsymbol{M}_2$. Thus by choosing $\varepsilon_1, \varepsilon_2$ to be of order $\mathcal{O}(\delta)$ we get that $\tilde{\boldsymbol{X}}_2 = \boldsymbol{X}_2^* + \dfrac{\delta}{C}\|M\|$ for some $C$ chosen to be large enough. Then for the last step consider the follow (sub-)rows of the matrix $\boldsymbol{X}$

$$
\begin{bmatrix}
\ell & * & * & \dots & * \\
\mathbf{r}_1 & \mathbf{r}_1 & \mathbf{r}_1 & \dots & \mathbf{r}_1 \\
\mathbf{r}_1 & \mathbf{r}_2 & \mathbf{r}_3 & \dots & \mathbf{r}_N
\end{bmatrix}
\tag{A.22}
$$

where $\mathbf{r}_i$ is the binary representation of position $i$. By choosing $\boldsymbol{W}_{\mathbf{K}}, \boldsymbol{W}_{\mathbf{Q}}$ as to

$$
\boldsymbol{W}_{\mathbf{K}} \boldsymbol{X} = \begin{bmatrix} \mathbf{r}_1 & \mathbf{r}_2 & \dots & \mathbf{r}_N \end{bmatrix}, \quad
\boldsymbol{W}_{\mathbf{Q}} \boldsymbol{X} = \begin{bmatrix} \mathbf{r}_1 & \mathbf{r}_1 & \dots & \mathbf{r}_1 \end{bmatrix}
\tag{A.23}
$$

---

[5] the size that we need will be $2 \log N_{\max} + 1$, where $N_{\max}$ is the maximum sequence length

[6] Note that the true error is even smaller since the operation performed only affects one row.

and consider $W_V X = \begin{bmatrix} \ell & * & \dots & * \\ 0 & 0 & \dots & 0 \end{bmatrix}$ we have that

$$attn(X) = X + W_V X \, softmax((W_K X)^\top W_Q X)$$

$$= X + \begin{bmatrix} \ell & * & \dots & * \\ 0 & 0 & \dots & 0 \end{bmatrix} softmax\left(\begin{bmatrix} r_1^\top r_1 & r_1^\top r_1 & \dots & r_1^\top r_1 \\ r_1^\top r_2 & r_1^\top r_2 & \dots & r_1^\top r_2 \\ \vdots & \vdots & \ddots & \vdots \\ r_1^\top r_N & r_1^\top r_N & \dots & r_1^\top r_N \end{bmatrix}\right)$$

$$= X + \begin{bmatrix} \ell & * & \dots & * \\ 0 & 0 & \dots & 0 \end{bmatrix} \begin{bmatrix} 1 & 1 & \dots & 1 \\ 0 & 0 & \dots & 0 \\ \vdots & \vdots & \dots & \vdots \\ 0 & 0 & \dots & 0 \end{bmatrix} + \varepsilon_3 M_3$$

By subtracting one identity head for the first that we focus on as described in Lemma 1 we have that $attn(X)$ results in the desired matrix. We output this result and overwrite $\ell/N$, thus we have

$$\begin{bmatrix} x_1 & \dots & s_1^{\ell-1} & s_1^\ell & \dots & s_2^{\ell-1} & s_2^\ell & \dots & s_n^L & x_{\text{test}} & \dots & \hat{s}^{\ell-1} \\ 0 & \dots & 0 & 0 & \dots & 0 & 0 & \dots & 0 & 0 & \dots & 0 \\ \hline 1 & \dots & \ell & \ell+1 & \dots & \ell & \ell+1 & \dots & L+1 & 1 & \dots & \ell \\ 1 & \dots & \ell & \ell+1 & \dots & \ell & \ell+1 & \dots & L+1 & 1 & \dots & \ell \\ \hline N & \dots & N & N & \dots & N & N & \dots & N & N & \dots & N \\ \ell & \dots & \ell & \ell & \dots & \ell & \ell & \dots & \ell & \ell & \dots & \ell \\ 0 & \dots & 0 & 0 & \dots & 0 & 0 & \dots & 0 & 1 & \dots & 1 \\ 1 & \dots & 1 & 1 & \dots & 1 & 1 & \dots & 1 & 1 & \dots & 1 \\ m_1 & \dots & m_\ell & m_{\ell+1} & \dots & m_{L+\ell+1} & m_{L+\ell+2} & \dots & m_{n(L+1)} & m_{n(L+1)+1} & \dots & m_N \end{bmatrix} \quad \text{(A.24)}$$

The new error introduced by this operation is again controlled as to be $\varepsilon_3 M_3 = \dfrac{\delta}{C}$, with $\|M_3\| \leq 1$.

**Step 3: Create $\ell+1$ (0 layer).** We first copy the row with $\ell$ to the row with $N$, this can trivially be done with a ReLU layer that outputs zero everywhere else except for the row of $N$s that output $(\ell)_+ - (N)_+$ to account also for the residual. We now use one of the bias terms (notice that $\ell$ is always positive) and set to one in one of the two rows that contain the $\ell$, again we account for the residual as before; everything else remains unchanged. Thus, we have

$$\begin{bmatrix} x_1 & \dots & s_1^{\ell-1} & s_1^\ell & \dots & s_2^{\ell-1} & s_2^\ell & \dots & s_n^L & x_{\text{test}} & \dots & \hat{s}^{\ell-1} \\ 0 & \dots & 0 & 0 & \dots & 0 & 0 & \dots & 0 & 0 & \dots & 0 \\ \hline 1 & \dots & \ell & \ell+1 & \dots & \ell & \ell+1 & \dots & L+1 & 1 & \dots & \ell \\ 1 & \dots & \ell & \ell+1 & \dots & \ell & \ell+1 & \dots & L+1 & 1 & \dots & \ell \\ \hline \ell+1 & \dots & \ell+1 & \ell+1 & \dots & \ell+1 & \ell+1 & \dots & \ell+1 & \ell+1 & \dots & \ell+1 \\ \ell & \dots & \ell & \ell & \dots & \ell & \ell & \dots & \ell & \ell & \dots & \ell \\ 0 & \dots & 0 & 0 & \dots & 0 & 0 & \dots & 0 & 1 & \dots & 1 \\ 1 & \dots & 1 & 1 & \dots & 1 & 1 & \dots & 1 & 1 & \dots & 1 \\ m_1 & \dots & m_\ell & m_{\ell+1} & \dots & m_{L+\ell+1} & m_{L+\ell+2} & \dots & m_{n(L+1)} & m_{n(L+1)+1} & \dots & m_N \end{bmatrix} \quad \text{(A.25)}$$

This operation can collectively be implemented (add the bias + copy the row) in the ReLU layer of the previous transformer layer that was not used in the previous step.

**Step 4: Create the binary bits (2 layers).** We will now use the information extracted in the previous steps, to create the binary indicators/bit to identify which tokens we want to filter. This can be easily implemented with one layer of transformer and especially the ReLU part of it. Notice that if we subtract the row that contains the tokens that have already been predicted, *i.e.*, $[\ell \, \ell \dots \ell]$ from the row that contains $[1 \, 2 \dots L \, 1 \, 2 \dots L \dots L]$ we will get zero only in the positions that we want to filter out and some non-zero quantity in the rest. This is trivially implemented with one ReLU layer. So, we need to implement an if..then type of operation. Basically, if the quantity at hand is zero we want to set the bit to one, while if it non-zero to set it to be zero. This can be implemented with the following ReLU part of a transformer layer

$$\mathrm{TF}(x_i) = 1 - (x_i)_+ - (-x_i)_+ + (x_i - 1)_+ + (-x_i - 1)_+ - ((x_i)_+ - (-x_i)_+) \qquad (A.26)$$

the last two terms are to account for the residual. Again the rest of the rows do not change and are zeroed-out.

**Step 5: Implement the filtering (1 layer).** We now apply Lemma 4 and our proof is completed. In this step as in Lemma 4, the error remains of the order of the error of the previous step. Thus, by fine-tuning the constants appropriately, depending on 1) the bound on the input as of Assumption 2, the targeted error $\delta$ and the constant used in Lemma 4 we achieve that the error has the target upped bound. ∎

**Theorem 2 (Theorem 1 restated)** *Consider a prompt $\boldsymbol{p}_n(f)$ generated from an L-layer MLP $f(\cdot)$ as described in Definition 1, and assume given test example $(\boldsymbol{x}_{test}, \boldsymbol{s}_{test}^1, \dots \boldsymbol{s}_{test}^L)$. For any resolution $\epsilon > 0$, there exists $\delta = \delta(\epsilon)$, iteration choice $T = \mathcal{O}(\kappa_{\max}^2 \log(1/\epsilon))$, and a backend transformer construction $TF_{BE}$ such that the concatenated transformer $TF = TF_{LR} \circ TF_{BE}$ implements the following: Let $(\hat{\boldsymbol{s}}^i)_{i=1}^{\ell-1}$ denote the first $\ell-1$ CoT-I/O outputs of TF and set $\boldsymbol{p}[\ell] = (\boldsymbol{p}_n(f), \boldsymbol{x}_{test}, \hat{\boldsymbol{s}}^1 \dots \hat{\boldsymbol{s}}^{\ell-1})$. At step $\ell$, TF implements*

1. ***Filtering.*** *Define the filtered prompt with input/output features of layer $\ell$,*

$$\boldsymbol{p}_n^{filter} = \begin{pmatrix} \dots \boldsymbol{0}, \, \boldsymbol{s}_1^{\ell-1}, \, \boldsymbol{0} \, \dots \boldsymbol{0}, \, \boldsymbol{s}_n^{\ell-1}, \, \boldsymbol{0} \, \dots \boldsymbol{0}, \, \hat{\boldsymbol{s}}^{\ell-1} \\ \dots \boldsymbol{0}, \quad \boldsymbol{0}, \quad \boldsymbol{s}_1^{\ell} \dots \boldsymbol{0}, \quad \boldsymbol{0}, \quad \boldsymbol{s}_n^{\ell} \dots \boldsymbol{0}, \quad \boldsymbol{0} \end{pmatrix}.$$

*There exists a fixed projection matrix $\boldsymbol{\Pi}$ that applies individually on tokens such that the backend output obeys $\|\boldsymbol{\Pi}(TF_{BE}(\boldsymbol{p}[\ell])) - \boldsymbol{p}_n^{filter}\| \leq \delta$.*

2. ***Gradient descent.*** *The combined model obeys $\|TF(\boldsymbol{p}[\ell]) - \boldsymbol{s}_{test}^{\ell}\| \leq \ell \cdot \epsilon / L$.*

*$TF_{BE}$ has constant number of layers independent of $T$ and $n$. Consequently, after $L$ rounds of CoT-I/O, TF outputs $f(\boldsymbol{x}_{test})$ up to $\epsilon$ accuracy.*

**Proof.** We apply Lemma 5 from which it is clear that there exists a projection such that the result stated in (1. Filtering) holds, and it is independent to $T$, $n$ and $\ell$. Next we turn to prove (2. Gradient descent). In Definition 1, we assume that the network's activation function is leaky-ReLU, *i.e.*,

$$\phi(x) = \begin{cases} x, & \text{if } x \geq 0 \\ \alpha x, & \text{otherwise.} \end{cases} \qquad (A.27)$$

Thus, as a first step we construct the inverse of leaky-ReLU and apply it in the second row of $\boldsymbol{p}_n^{\text{filter}}$ where the inverse of leaky-ReLU is

$$\phi^{-1}(y) = \begin{cases} y, & \text{if } y \geq 0 \\ y/\alpha, & \text{otherwise.} \end{cases} \qquad (A.28)$$

This can be implemented with the following activation function (denoted by $\sigma(\cdot)$) using ReLUs:

$$\sigma(x) = (x)_+ - 1/\alpha(-x)_+. \qquad (A.29)$$

After it, it remains $\mathrm{TF}_{\mathrm{LR}}$ to solve linear regression problems. Taking $\ell$th layer, first neuron prediction as an example, and letting $\boldsymbol{x}_i' := \boldsymbol{s}_i^{\ell-1}$, $y_i' := \phi^{-1}(\boldsymbol{s}_i^\ell[0])$ and $\boldsymbol{w} = \boldsymbol{W}_\ell[0]$, linear regression has form of $y_i' = \boldsymbol{w}^\top \boldsymbol{x}_i'$ for $i \in [n]$. Notice that the extra zeros do not contribute in the update performed by gradient descent, and after gradient descent has been performed, we apply back the leaky ReLU. Then following Assumption 1, since we assume $\mathrm{TF}_{\mathrm{LR}}$ performs the same as gradient descent optimizer,

given matrix condition as described in Definition 1, after running $T$ iterations of gradient descend on the linear regression problem and considering each layer prediction with resolution $\epsilon/L$, we can get that $\|\text{TF}_{\text{LR}}(\boldsymbol{p}[\ell]) - \bar{\boldsymbol{s}}^\ell\| \leq \epsilon/L$, where $\bar{\boldsymbol{s}}^\ell = \phi(\boldsymbol{W}_\ell \hat{\boldsymbol{s}}^{\ell-1})$ is the correct prediction if taking $\hat{\boldsymbol{s}}^{\ell-1}$ as input. Then we have

$$\|\text{TF}_{\text{LR}}(\boldsymbol{p}[\ell] - \boldsymbol{s}^\ell_{\text{test}})\| \leq \|\text{TF}_{\text{LR}}(\boldsymbol{p}[\ell]) - \bar{\boldsymbol{s}}^\ell\| + \|\boldsymbol{W}_\ell(\hat{\boldsymbol{s}}^{\ell-1} - \boldsymbol{s}^{\ell-1}_{\text{test}})\| \lesssim \epsilon/L + \|\text{TF}_{\text{LR}}(\boldsymbol{p}[\ell-1] - \boldsymbol{s}^{\ell-1}_{\text{test}})\|.$$

Let $\text{TF}_{\text{LR}}(\boldsymbol{p}[0])$ returns $\boldsymbol{x}_{\text{test}}$ and therefore $\|\text{TF}_{\text{LR}}(\boldsymbol{p}[0]) - \boldsymbol{x}_{\text{test}}\| = 0$. Combing results in that $\|\text{TF}_{\text{LR}}(\boldsymbol{p}[\ell] - \boldsymbol{s}^\ell_{\text{test}})\| \lesssim \ell \cdot \epsilon/L$. Since from Lemma 5 we have that we can choose $\delta$ to be arbitrary. Let $\delta = \epsilon/L$, where $L$ is the total predictions we will make. Then it will result in $\|\text{TF}(\boldsymbol{p}[\ell] - \boldsymbol{s}^\ell_{\text{test}})\| \lesssim \ell \cdot \epsilon/L$, which completes the proof.

∎

# B    Experimental Details

In this section, we provide the implementation details of our experiments.

## B.1    Model Evaluation

Recap the same setting as in Section 2.3 and assume we have pretrained models with parameters $\hat{\boldsymbol{\theta}}^{\text{CoT-I}}$ and $\hat{\boldsymbol{\theta}}^{\text{CoT-I/O}}$. Next we make predictions following Section 2.2. Letting $\ell(\cdot, \cdot) : \mathcal{Y} \times \mathcal{Y} \to \mathbb{R}$ be loss function, we can define test risks as follows.

$$\mathcal{L}^{\text{CoT-I}}(n) = \mathbb{E}_{(\boldsymbol{x}_i)_{i=1}^n, (f_\ell)_{\ell=1}^L} \left[ \ell(\hat{\boldsymbol{y}}_n, f(\boldsymbol{x}_n)) \right] \quad \text{where} \quad \hat{\boldsymbol{y}}_n = \text{TF}(\boldsymbol{p}_n(f), \boldsymbol{x}_n; \hat{\boldsymbol{\theta}}^{\text{CoT-I}})$$

and

$$\mathcal{L}^{\text{CoT-I/O}}(n) = \mathbb{E}_{(\boldsymbol{x}_i)_{i=1}^n, (f_\ell)_{\ell=1}^L} \left[ \ell(\hat{\boldsymbol{y}}_n, f(\boldsymbol{x}_n)) \right] \quad \text{where} \quad \hat{\boldsymbol{y}}_n = \text{TF}(\boldsymbol{p}_n(f), \boldsymbol{x}_n, \hat{\boldsymbol{s}}^1 \cdots, \hat{\boldsymbol{s}}^{L-1}; \hat{\boldsymbol{\theta}}^{\text{CoT-I/O}}).$$

Here, we use $\mathcal{L}(n)$ to define the test risk when given prompt with $n$ in-context samples. Then, results shown in Figures 3&4&5&6&7(a) are test risks $\mathcal{L}(n)$ given $n \in [N]$. Following model training and evaluation, we can see that once loss function is the same for both training and predicting, CoT-I (as well as ICL) accepts training risk $\mathcal{L}^{\text{CoT-I}}_{\text{train}} = \frac{1}{N} \sum_{n=1}^N \mathcal{L}^{\text{CoT-I}}(n)$.

## B.2    Implementation

All the transformer experiments use the GPT-2 model [Radford et al., 2019] and our codebase is based on prior works [Garg et al., 2022, Wolf et al., 2019]. Specifically, the model is trained using the `Adam` optimizer with learning rate $0.0001$ and batch size $64$, and we train $500\text{k}$ iterations in total for all ICL, CoT-I and CoT-I/O methods. Each iteration randomly samples inputs $\boldsymbol{x}$s and functions $f$s. We also apply curriculum learning over the prompt $n$ as Garg et al. [2022] did for 2-layer random MLPs (Sec. 4.2). For both training and testing, we use the squared error as the loss function, *i.e.*, $\ell(\hat{\boldsymbol{y}}, \boldsymbol{y}) = \|\hat{\boldsymbol{y}} - \boldsymbol{y}\|^2$ (or $\ell(\hat{y}, y) = (\hat{y} - y)^2$ for scalar).

# C    Additional Experimental Results

## C.1    Out-of-distribution Experiments

While our primary focus lies in in-distribution scenarios, wherein the test examples are presumed to follow the same distribution as the training data, this subsection extends our analysis to out-of-distribution (OOD) settings.

To provide a clearer understanding of our model's behavior under distribution shifts and to quantitatively assess the impact on test risk, we conduct experiments with varying levels of distribution shift. Specifically, we evaluate the test risk when the prompt is supplemented with 100 in-context examples. The results of these experiments are depicted in Figure 9, which serves to illustrate how our model's performance is affected under these OOD conditions. In Fig. 9(a), we analyze noisy in-context samples during testing. The solid and dashed curves represent the test risks, corresponding to the noisy in-context samples whose (input, output) takes the form of either $(\boldsymbol{x}, y + \text{noise})$ or

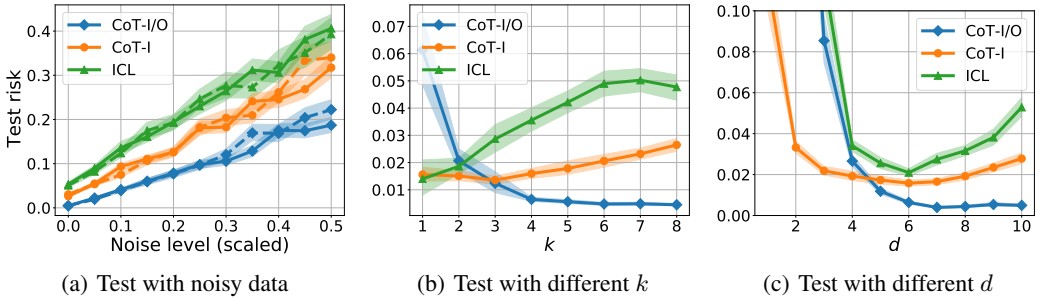

(a) Test with noisy data      (b) Test with different $k$      (c) Test with different $d$

Figure 9: We implement robustness experiments under different distribution shift levels. In Fig. 9(a) we add noise to the label $y$ (solid) or input features $\boldsymbol{x}$ (dashed). In Fig. 9(b), we in-context learn an MLP with $k \in [8]$ hidden nodes whereas transformer is trained for MLPs with 8 hidden nodes. In Fig. 9(b), we consider misspecification of input dimension: TF is trained with $d = 10$ but we feed a neural net with $d \leq 10$.

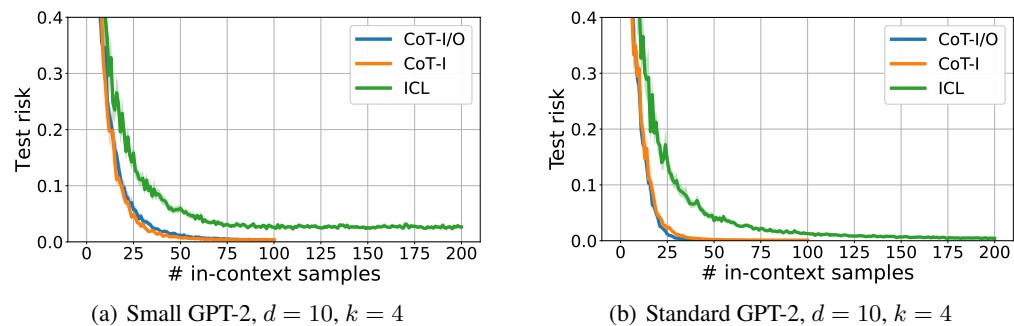

(a) Small GPT-2, $d = 10, k = 4$      (b) Standard GPT-2, $d = 10, k = 4$

Figure 10: We train ICL with more in-context examples. The main conclusion is that: For small GPT, ICL can indeed not approximate the neural net even with many examples (unlike CoT) whereas, for large GPT, ICL can do so (although much less efficient). This is in line with our theoretical intuitions on the expressivity benefits of CoT.

$(\boldsymbol{x} + \text{noise}, y)$, respectively. The results indicate that CoT exhibits greater robustness compared to ICL, and the test risks increase linearly with the noise level, with attributed to the randomized MLPs setting. Additionally, in Figs. 9(b)&9(c), we instead explore out-of-distribution test tasks where test MLPs differ in $(d, k)$ from the training phase. For both subfigures, we firstly train small GPT-2 using 2-layer MLPs with $d = 10, k = 8$. In Fig. 9(b), we fix $d = 10$ and vary $k$ from 1 to 8, whereas in Fig. 9(c), we fix $k = 8$ and vary $d$ from 1 to 10. In both instances, the findings reveal that CoT's performance remains almost consistent when $k \geq 4$ or $d \geq 6$, and ICL is unable to surpass it. The improved performance of ICL with smaller values of $d$ or $k$ again reinforces our central assertion that ICL requires $O(dk)$ samples for in-context learning of the 2-layer random MLP, and reducing either $d$ or $k$ helps in improving the performance. Given that we employ the ReLU activation function, smaller values of $d$ or $k$ can lead to significant bias in the intermediate feature. Consequently, CoT cannot derive substantial benefits from this scenario, resulting in a decline in performance.

## C.2 Further Evidence of Model Expressivity

There are two determinants of model learnability in in-context tasks: sample complexity and model expressivity. Sample complexity pertains to the number of samples needed to precisely solve a problem. However, when the transformer model is small, even with a sufficiently large number of samples, due to its lack of expressivity, ICL cannot achieve zero test risk. This contrasts with CoT, which decomposes complex tasks into simpler sub-tasks, thereby requiring smaller models for expression. Figure 5 has illustrated the expressivity of different GPT-2 architectures, showing that the tiny GPT-2 model is too small to express even a single layer of 2-layer MLPs. Additionally, we have run more experiments, and the results are shown in Figure 10. Both Figures 10(a) and 10(b)

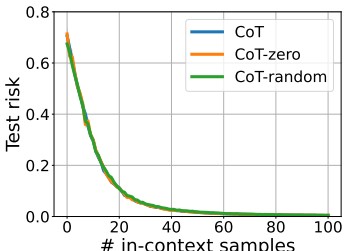

(a) Test filtering on first layer prediction

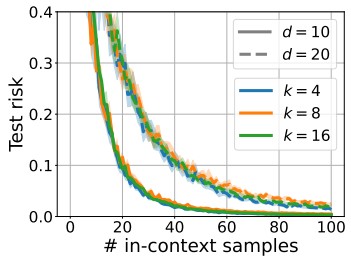

(b) CoT-I/O: composed risk (same as Fig. 3(a))

Figure 11: Fig. 11(a) presents a filtering evidence of 2-layer MLPs. Given a 2-layer MLP in-context example $(\boldsymbol{x}, \boldsymbol{s}, y)$, CoT admits $(\boldsymbol{x}, \boldsymbol{s}, y)$ as test sample; while test samples of CoT-zero and CoT-random are formed by $(\boldsymbol{x}, \boldsymbol{s}, 0)$ and $(\boldsymbol{x}, \boldsymbol{s}, z)$ where $z \sim \mathcal{N}(0, d)$. Fig. 11(b) is directly cloned form Fig. 3(a) with error bar for better comparison with results in Fig. 12.

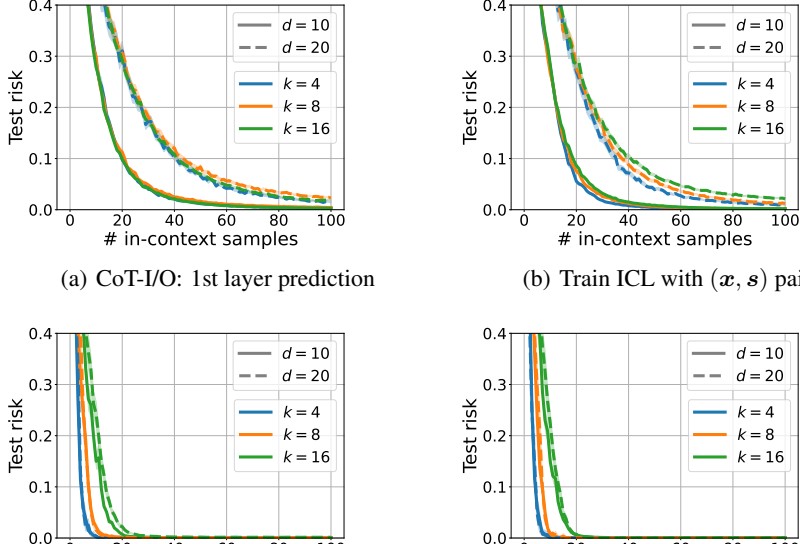

(a) CoT-I/O: 1st layer prediction

(b) Train ICL with $(\boldsymbol{x}, \boldsymbol{s})$ pairs

(c) CoT-I/O: 2nd layer prediction

(d) Train ICL with $(\boldsymbol{s}, y)$ pairs

Figure 12: We compare the performance of filtered CoT and ICL. In Fig. 12(a)&12(c), we decouple the composed risk of predicting 2-layer MLPs into risks of individual layers (following Section 4.1), which shows the filtered CoT results. In Fig. 12(b)&12(d), we train two additional models using ICL method taking $(\boldsymbol{x}, \boldsymbol{s})$ and $(\boldsymbol{s}, y)$ as inputs.

detail training models with MLP tasks of dimensions $d = 10$ and $k = 4$. In Fig. 10(a), we use a small GPT-2 model, and the results show that the test risk stops decreasing even with more in-context examples. In Fig. 10(b), we train a larger model, and the results demonstrate that the standard GPT-2 is sufficient to express a 2-layer MLP with $d = 10$ and $k = 4$.

## C.3 Filtering Evidence in 2-layer MLPs

Section 5 has demonstrated the occurrence of filtering in the linear deep MLPs setting (black dotted curves in Fig. 8(b)). In this section, we present further empirical evidence based on the 2-layer MLPs setting discussed in Section 4.2.

Follow the same setting as Figure 3(a) and choose $d = 10$ and $k = 8$. Assume we have a model pretrained using CoT-I/O method. As described in Section 4.2, during training, the prompt consists of in-context samples in the form of $(\boldsymbol{x}, \boldsymbol{s}, y)$ where $\boldsymbol{s} = (\boldsymbol{W}\boldsymbol{x})_+$ and $y = \boldsymbol{v}^\top \boldsymbol{s}$. To investigate filtering,

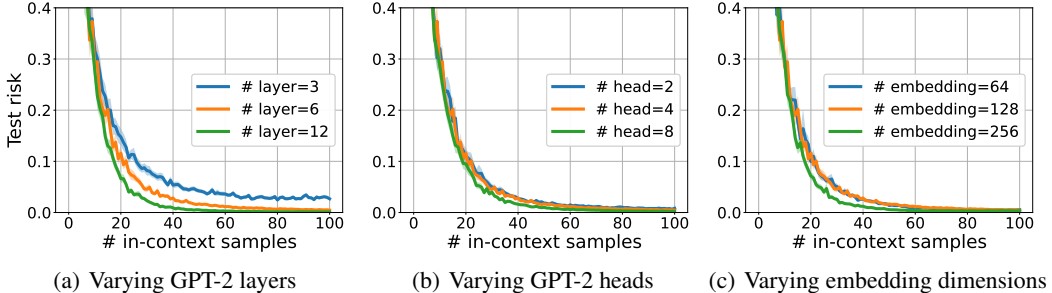

|   (a) Varying GPT-2 layers   |   (b) Varying GPT-2 heads   |   (c) Varying embedding dimensions   |

Figure 13: To further investigate how model architectures impact the prediction performance, we fix the number of heads and embedding dimension in Fig. 13(a) and change the layer number in $\{3, 6, 12\}$. Similarly for Fig. 13(b)&13(c) but instead, change number of heads (in $\{2, 4, 8\}$) and embedding dimensions (in $\{64, 128, 256\}$).

we make three different predictions to evaluate the intermediate output, whose test prompts have in-context examples with the following forms:

$$\text{CoT: } (\boldsymbol{x}, \boldsymbol{s}, y), \quad \text{CoT-zero: } (\boldsymbol{x}, \boldsymbol{s}, 0), \quad \text{CoT-random: } (\boldsymbol{x}, \boldsymbol{s}, z),$$

where $z \sim \mathcal{N}(0, d)$. The results are displayed in Figure 11(a) where blue, orange and green curves represent first layer prediction results using CoT, CoT-zero and CoT-random prompts, respectively. From this figure, we observe that the three curves are well aligned, indicating that when making a prediction for input $\boldsymbol{x}$, TF will attend only to $(\boldsymbol{x}, \boldsymbol{s})$ and ignore $y$. Therefore filling the positions of $y$ with any random values (or zero) will not change the performance of first layer prediction.

### C.4 Comparison of Filtered CoT with ICL

Until now, many experimental results have shown that CoT-I/O provides benefits in terms of sample complexity and model expressivity compared to ICL. As an interpretation, we state that CoT can be decoupled into two phases: *Filtering* and *ICL*, and theoretical results have been provided to prove this statement. As for the empirical evidence, Sections 5 and C.3 precisely show that filtering does occur in practice. In this section, we provide additional experiments to demonstrate that, after filtering, CoT performs similarly to ICL.

For convenience and easier comparison, we repeat the same results as Fig. 3(a) in Fig. 11(b), where $d \in \{10, 20\}$, $k \in \{4, 8, 16\}$, and train with a small GPT-2. We again recap the data setting for the 2-layer MLP, where the in-context examples of CoT prompt are in the form of $(\boldsymbol{x}, \boldsymbol{s}, y)$. Given that filtering happens, we make first and second layer predictions following Section 4.1 and results are presented in Fig. 12(a) and Fig. 12(c), respectively. These results show the performances of the filtered CoT prompts. Next, we need to compare the performance with separate ICL training. To achieve this goal, we train a small GPT-2 model using ICL method with prompt containing $(\boldsymbol{x}, \boldsymbol{s})$ pairs (first layer). The test results are shown in Fig. 12(b). Additionally, in Fig. 12(d), we train another small GPT-2 model using ICL but with prompts containing $(\boldsymbol{s}, y)$ pairs (second layer). By comparing Fig. 12(a) and 12(b), as well as Fig. 12(c) and 12(d), we observe that after filtering, CoT-I/O achieves similar performance as individually training a single-step problem through the ICL phase.

### C.5 CoT across Different Sizes of GPT-2

In Figure 5, we have demonstrated that larger models help in improving performance due to their ability of solving more complex function sets. However, since tiny, small and standard GPT-2 models scale the layer number, head number and embedding dimension simultaneously, it is difficult to determine which component has the greatest impact on performance. Therefore in this section, we investigate how different components of transformer model affect the resulting performance by run CoT-I/O on various GPT-2 architectures.

We maintain the same setting as in Section 4.2, fix $d = 10$ and $k = 8$, and consider a base GPT-2 model (small GPT-2) with 6 attention layers, 4 heads in each layer and 128-dimensional embeddings.

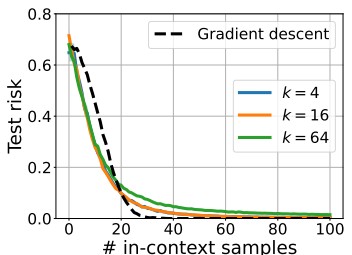
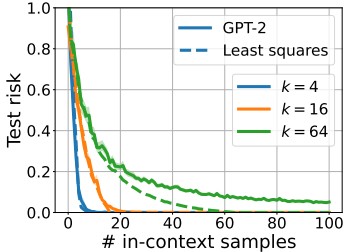

(a) 1st layer: compare with GD solving ReLU    (b) 2nd layer: compare with least squares

Figure 14: Fig. 14(a): compare transformer results (solid) with gradient descent optimizer (dashed) when solving first layer of 2-layer MLPs; Fig. 14(b): compare transformer result (solid) with least squares optimizer (dashed) when solving the second layer of 2-layer MLPs.

In Fig. 13(a), we fix the number of heads at 4 and the embedding dimension at 128, while varying the number of layers in $\{3, 6, 12\}$. Similarly, we explore different models with different numbers of heads and embedding dimensions, and the results are respectively presented in Fig. 13(b) and 13(c). Comparing them, we can observe the following: 1) once the problem is sufficiently solved, increasing the model size does not significantly improve the prediction performance (see Fig. 13(b)&13(c)); 2) the number of layers influences model expressivity, particularly for small GPT-2 architecture (see Fig. 13(a)).

### C.6 Comparison of Transformer Prediction and Linear Regression

We also provide experimental findings to verify Assumption 1 in this section. Previous work [Giannou et al., 2023, Akyürek et al., 2022] has theoretically proven that TF can perform similar to gradient descent, and empirical evidence from [Dai et al., 2022, Garg et al., 2022, Li et al., 2023c] suggests that TF can even be competitive with Bayes optimizer in certain scenario. To this end, we first repeat the same first/layer predictions from Figure 4 in Figure 14, where $d = 10$ and blue, orange and green solid curves represent the performances of $k = 4, 16, 64$ using pretrained small GPT-2 models. We also display the evaluations of gradient descent/least square solutions in dashed curves. Specifically, in Fig. 14(a) , we solve problem

$$\hat{\boldsymbol{w}}_n = \arg\min_{\boldsymbol{w}} \frac{1}{n} \sum_{i=1}^{n} \|(\boldsymbol{w}^\top \boldsymbol{x}_i)_+ - y_i\|^2 \text{ where } \boldsymbol{x}_i \sim \mathcal{N}(0, \boldsymbol{I}_d), \ y_i = (\boldsymbol{w}^{\star\top} \boldsymbol{x}_i)_+$$

for some $\boldsymbol{w}^\star \sim \mathcal{N}(0, 2\boldsymbol{I}_d)$ and $n$ is the training sample size. Then the normalized test risks are computed by $\mathcal{L}(n) = \mathbb{E}_{\boldsymbol{w}^\star, \boldsymbol{x}}[\|(\hat{\boldsymbol{w}}_n^\top \boldsymbol{x})_+ - y\|^2]/d$, and we show point-to-point results for $n \in [N]$ in black dashed curve in Fig. 14(a)[7]. As for the second layer, we solve least squares problems as follows

$$\hat{\boldsymbol{v}}_n = \boldsymbol{S}^\dagger \boldsymbol{y} \text{ where } \boldsymbol{S} \in \mathbb{R}^{n \times k}, \ \boldsymbol{S}[i] = (\boldsymbol{W}^\star \boldsymbol{x}_i)_+, \ \boldsymbol{y}[i] = \boldsymbol{v}^{\star\top} \boldsymbol{S}[i], \ \boldsymbol{x}_i \in \mathcal{N}(0, \boldsymbol{I}_d)$$

for some $\boldsymbol{W}^\star \in \mathbb{R}^{k \times d} \sim \mathcal{N}(0, 2/k)$ and $\boldsymbol{v}^\star \sim \mathcal{N}(0, \boldsymbol{I}_k)$. Here, $\dagger$ represents the pseudo-inverse operator. Then we calculate the normalized test risk of least square solution (given $n$ training samples) as $\mathcal{L}(n) = \mathbb{E}_{\boldsymbol{W}^\star, \boldsymbol{v}^\star, \boldsymbol{x}}[\|\hat{\boldsymbol{v}}_n^\top \boldsymbol{s} - y\|^2]/d$ where $\boldsymbol{s} = (\boldsymbol{W}^\star \boldsymbol{x})_+$ and $y = \boldsymbol{v}^{\star\top} \boldsymbol{s}$. The results are presented in Fig. 14(b) where blue, orange and green dashed curves correspond to solving the problem using different values of $k \in \{4, 16, 64\}$. In this figure, the curves for $k = 4, 16$ are aligned with GPT-2 risk curves, which indicates that TF can efficiently solve linear regression as a least squares optimizer. However, the curve for $k = 64$ does not align, which can be attributed to the increased complexity of the function set with higher dimensionality ($k = 64$). Learning such complex functions requires a larger TF model.

---

[7]To mitigate the bias introduced by ReLU activation, we subtract the mean value during prediction, *i.e.*, $\mathcal{L}(n) = \mathbb{E}_{\boldsymbol{w}^\star, \boldsymbol{x}}[\|(\hat{\boldsymbol{w}}_n^\top \boldsymbol{x})_+ - y - (\mathbb{E}_{\boldsymbol{x}}[(\hat{\boldsymbol{w}}_n^\top \boldsymbol{x})_+ - y])\|^2]/d$.

