# OpenReview forum: "Dissecting Chain-of-Thought: Compositionality through In-Context Filtering and Learning"
_NeurIPS.cc/2023/Conference — NeurIPS 2023 poster_

### Official Review · Reviewer_wZge · 2023-06-22

**Soundness:** 3 good
**Presentation:** 1 poor
**Contribution:** 3 good
**Rating:** 6
**Confidence:** 3

**Summary:**

This paper investigates how the transformer-based model can compositionally learn some complex functions (e.g., an MLP) by breaking them down to some atom problems (e.g., linear mapping). Such an ability is also the crux of the success of Chain-of-thought (CoT) in-context learning (ICL) methods in large language models. Hence this paper bridges the gap between these two fields by theoretically analyzing the sample complexity of different methods and the accelerating effect of CoT in pertaining. Although I believe the contribution of the paper is solid, novel, and helpful to the field, there are plenty of problems that make the paper hard to follow. I wish the authors could polish the paper and tackle some of my concerns to make the paper stronger. Hence at this stage, I would only give a borderline rejection. I would be very happy to increase my score during the rebuttal phase.

**Strengths:**

See the summary part.

**Weaknesses:**

1. I find the paper hard to follow, maybe because the theoretical part, which I believe is the most important contribution, is too abstract. It is not easy to get intuition from current section 3.2, so maybe explaining how theorem 1 is formulated is helpful.

2. The experimental parts are not well organized. First, it appears almost everywhere (in section 2,3,4), which breaks the flow of the paper significantly. Also, some of the subfigures in Fig3,4,5 contain similar information. So it would be good to re-organize these results to make the paper easier to follow.

3. In section 4.3, the paper switches from the ICL setting to the pertaining setting. It is beneficial to clearly explain what’s the difference between them and why we need experiments in this setting.

**Questions:**

1. For the experiment in Figure 1, will ICL and CoT-I converge to a lower value if the number of samples keeps increasing? Because the sample complexity of CoT-I/O is lower than the other two and it converges at around 100 samples, would the other two methods converge if we use 500 samples?

2. In Figure 2,3, it is hard to distinguish different methods using dashed lines, using different markers might be good.

3. In Figure 3, if we want to see the influence of k, why not draw a figure using k as the x-axis?

4. Figure 3 is on page 2, but it is first referred at page 4 — it is hard to locate the figure when reading the paper.

5. Some typos and imprecise expressions:

    a.) In section 3.1, the first sentence, ‘we train 2-layer MLPs’. Do we train this MLP? IIUC, this MLP is used to generate the training samples and is fixed all the time.

    b.) In section 3.2, the last paragraph, ‘consider the consider the condition’.

**Limitations:**

I’m not sure whether this is an unavoidable problem of the CoT-style ICL, but it would be nice if the author can answer the following problem somewhere in the paper, which might make the paper more solid. I think comparing the sample complexity (i.e., the number of examples in prompting) between these 3 algorithms is unfair. Imagine we have $k$ examples and the problem has L steps. Then in ICL, there are only $k$ input vectors and 1 supervisory signal. In CoT-I, there are $Lk$ input vectors and 1 supervisory signal. For CoT-I/O, there are $Lk$ input vectors and $(L-1)k$ supervisory signals (I’m not sure in this paper's setting, whether ground truth $s_n^l$ is accessible in CoT-I/O, but if that is true, the comparison would be more unfair.) Given the differences of the input and supervisory signals, it might be straightforward to conclude that CoT has smaller sample complexity than ICL.

---

> ### Author Rebuttal · Authors · 2023-08-10
>
> We thank the reviewer for their detailed feedback and encouraging feedback. We hope that we have addressed your questions and concerns adequately below.
>
> > **W1. Theory is too abstract:** We provide below some more detailed description of the construction that implements this filtering, which we will ensure to include in our revised version.
> >
> > First off, we include a figure in the uploaded pdf which shows the process in a high level depiction. The idea is to filter out pairs of data that are relevant to the current prediction. So, assuming that we currently want to predict the $l$-th layer's output. In that case we want to filter out the input data $x_{l-1},x_{l}$, $x_{L+l-},x_{L+l}, ...$, where $L$ is the total number of layers and $x_i$ are the input data. To do so, we implement the following steps:
> >
> >1. Given $n$ bits $b_i$, which take values $0$ or $1$ we zero out any data point $x_i$ that its corresponding bit is zero.
> 2. The second step is to construct these bits, which are the indicators of what needs to be filtered. As we also mention in our main paper, this procedure of filtering is agnostic to the token-to-be predicted and it is implemented in an automated way.
> >
> > A detailed proof is given in section A of the appendix. Finally, we will highlight that the *self-attention mechanism plays a critical role in both filtering and ICL stages highlighting the transformer-specific nature of our theorem*. For instance, attention layer is crucial for selecting which tokens should be processed next (see the attached Figure 1 for visualization). We will provide a discussion on this and also (experimentally) visualize the theorem's message on "Chain-of-Thought <=> Filtering + ICL + Looping" via attention maps.
>
> > **W2. Reorganize experimental section:** We recognize the lack of organization in our work and have made plans to revise and restructure it. Specifically, we will combine the experimental results and focus Section 3 purely on theory. Details can be found in the general response.
>
> > **W3. ICL vs Pretraining setting** Thanks for raising this question: The initial part of the paper focuses on the inference phase of in-context learning. That is, we are interested in the number of examples in the prompt to correctly predict the test query. By "pretraining" we mean the training phase of the in-context learning and we study the number of prompts the transformer needs to be trained with so that it can successfully in-context learn during inference. Our conclusion (based on deep linear MLPs) is that CoT helps improve the sample complexity of this training phase by learning shortcuts to represent complex functions. To avoid confusion, we will replace all "pretraining" phrases with "ICL training"
>
> > **Q1. Will ICL converge given more in-context samples? (attached Figure 5)** We appreciate the reviewer's query and the short answer is: NO (unless we enlarge the model size). There are two determinants of model learnability in in-context tasks: sample complexity and model expressivity. Sample complexity pertains to the number of samples needed to precisely solve a problem. However, when the transformer model is small, even with a sufficiently large number of samples, due to its lack of expressivity, ICL cannot achieve zero test risk. This contrasts with CoT, which decomposes complex tasks into simpler sub-tasks, thereby requiring smaller models for expression. Figure 4 in the paper illustrates the expressivity of different GPT-2 architectures, showing that the tiny GPT-2 model is too small to express even a single layer of 2-layer MLPs. Additionally, we have run more experiments, and the results are shown in Figure 5 in the attached file. Both Figures 5(a) and 5(b) detail training models with MLP tasks of dimensions $d=10$ and $k=4$. In Figure 5(a), we use a small GPT-2 model, and the results show that the test risk stops decreasing even with more in-context examples. In Figure 5(b), we train a larger model, and the results demonstrate that the standard GPT-2 is sufficient to express a 2-layer MLP with $d=10$ and $k=4$.
>
> > **Q2.** Thank you for the constructive point and suggestion. We will integrate the changes in the updated version.
>
> > **Q3.** We appreciate this perspective. We did consider this during our research. However, since test risks fluctuate significantly with in-context sample sizes, plotting the x-axis as $k$ presents challenges. We would have to either focus on a specific in-context sample size or display the averaged risks, making it difficult to determine whether CoT is universally superior to ICL or only in particular cases of in-context samples. What's more, current plots could clearly show the performance alignments over different settings. Nevertheless, we still wanted to illustrate how the test risks vary with $k$, and for this purpose, we have included a bar figure in Figure 5(a) in the paper, showing the averaged risks.
>
> > **Q4.** We apologize for the lack of organization and appreciate your patience. We have devised a plan to reorganize our work, as outlined in the general response, and we hope this addresses your concerns.
>
> > **Q5.** Thank you for pointing out the errors, and we apologize for any confusion. The sentences and typos have been corrected.
>
> > **Limitations:** We apprepriate the reviewer's thoughtful feedback. While we agree ICL has a smaller in-context window size compared to CoT, we want to emphasize that our setting is aligned with the common practice where reasoning prompts are typically denser than merely providing final answers, and explanations prove helpful. Moreover, as Reviewer zLZS syggests, we have conducted varant ICL experiments by filling the explanation with some random data and the results are shown in attached Figures 4. Fig 5(a) further illustrates that even with more in-context samples, ICL cannot surpass CoT due to task complexity and model expressivity limitations.

---

> > ### Comment · Reviewer_wZge · 2023-08-16
> > **Thanks for the feedback.**
> >
> > Thanks very much for the authors' feedback, which solves most of my concerns. As I mentioned in my original review, I believe the contribution of this paper is valuable and inspiring. Considering the author will improve the presentation in the next version, I would increase my score from 4 to 6. I am looking forward to see the final version.

---

> > > ### Author Response · Authors · 2023-08-17
> > > **Thank you!**
> > >
> > > Many thanks for your thoughtful review and positive feedback! Your suggestions have been very valuable, we have revised and further improved our presentation based on them.

---

### Official Review · Reviewer_zLZS · 2023-06-29

**Soundness:** 3 good
**Presentation:** 2 fair
**Contribution:** 3 good
**Rating:** 7
**Confidence:** 4

**Summary:**

The paper proposes to study chain-of-thought (prompting) in the setting of learning MLPs. The authors build on top of recent work studying in-context learning linear regression tasks in the light of gradient descent and extent their setting to learning non-linear functions. In order to study chain-of-thought prompting they either provide features / hidden activations from the "teacher" MLP the student (CoT-I setting) or make the Transformer produce its own input by "looping" (CoT-I/O  setting).
They provide theoretical results and empirical evidence that the Transformer is more sample efficient due to CoT by leveraging the given inputs and/or can remember weights coming from the family of teachers to allow for the self-production of features due to looping.


**Strengths:**

I like the abstraction of CoT presented in the paper. The experiments are convincing and supported by the theoretical results. Although this setting is quite simple, it nicely extends the recent studies of gradient descent on linear regression. The results and the line of thinking is very intuitive, I like the ideas and the empirical execution of the paper.

**Weaknesses:**

Although I am very familiar with the setup of the paper, I still had problems understanding it. The presentation can be made much clearer.
Please work on your presentation and think about how to structure the paper in a clearer and structured way. It is a bit confusing that you have 1.5 empirical sections and the paper is very dense. I think it would benefit the paper if the authors work on restructuring the presentation given the (hopefully available) extra page if accepted.

It would be nice to clearly explain what is meant with "learning an MLP". This differs from classic student teacher frameworks since you are not learning and also don't have to learn (in the CoT-I setting as far as I understand) the weight of the MLPs but solely rely on gradient descent on a linear regression task which acts on the given features in-context. Given these features in-context, it feels quite obvious that the CoT setting is indeed outperforming classic in-context learning. Nevertheless, I think it is still quite interesting.

Also it would be helpful to clearly state your abstraction/hypotheses that in LLMs the additional data needed to outperform plain ICL are "features" of the data motivating your MLP abstraction. This only became clear to me after reading a few times.

In the CoT-I/O setting, please explain why you are not using MLPs but linear deep nets as the teacher.  Although the presented experiments make sense, the training of Transformers is usually done differently as TFs are not trained to do CoT-I or CoT-I/O. Please comment on this.

There are a couple of ablations / interesting experiments I would like to see.

**Questions:**

My main concern is the following: Given that CoT is given the activations of the teacher, the sequence length of these experiments is larger. I think it is quite crucial to contrast the performance of your trained models when given different input data. If I understand correctly, you are hypothesizing that CoT works in LLMs is because the extra data is representative of features of the input data.

3 Naive ablations come to mind: Just train a model on your setup where the extra data that is provided during CoT (s_0, ...,) is either constant, random or actually the same data given multiple times. That would provide more evidence that in your setting indeed the teachers activations are crucial. It could (I doubt it myself) be that just more naive prompts do the trick.

**Limitations:**

I think the limitations that the setting is quite constructed and might not have anything to do how CoT works in LLMs could be a bit more strong.

---

> ### Author Rebuttal · Authors · 2023-08-10
>
> We thank the reviewer for their detailed review and encouraging feedback! We hope that we have addressed your questions and concerns adequately below.
>
> > **Lack of clarity:** We are sorry that the reviewer feels this way. Taking the reviewer's suggestion into sincere consideration, we are working on restructuring the paper to improve readability and hope that the new manuscript will be much better structured and easier to read. We have discussed our plan for reorganization in the general response. Basically, we will keep Section 3 pure theory, pack all the 2-layer MLP experiments (current Sections 3.1 and 4.2) together in Section 4, and Section 5 will further introduce our deep linear MLP results. We are glad to hear back if you have any further suggestions and comments.
>
> >**Explain "Learning an MLP":** We are encouraged that the reviewer finds our work interesting! In our work, we define "learning an MLP" as in-context learning a function whose performance is close to the target MLP. This is indeed as the reviewer comments, that transformers have the ability to implicitly learn the target MLP from the demonstrations without altering the model parameters. We will clarify this further in our updated manuscript. We also appreciate the reviewer's comment on the implicit gradient descent ability of ICL. Although the result that CoT > ICL is intuitive and not surprising, to the best of our knowledge, seldom do works look into its mechanism. Our work introduces a novel framework to explain it as CoT = filtering + ICL. We also provide more discussion regarding this in the general response and visualize it in attached Figure 1. Specifically, CoT prompts are filtered out into different steps of the ICL process, and the model performs gradient descent for each ICL and links the output sequentially. Theoretical and empirical evidence in this paper has been provided to prove it. To reiterate, our goal is to better understand an existing phenomenon rather than propose a new one.
>
> > **Clarify the connection to LLMs:** Thank you for the suggestion. We will definitely add more discussion regarding how our CoT setup can be related to real NLP settings in the revised manuscript.
>
> > **Why using deep linear MLPs instead of standard MLPs?** Great question! We could have indeed used standard MLPs. However, a drawback of ReLU activation is that, even with proper normalization, the feature distribution becomes more and more heavy tailed as we get deeper in the network. This heavy tail impedes the training process of the transformer resulting in very slow and brittle experimentation when depth is large. We opted to use deep linear MLPs as they better preserve the input distribution (especially with random unitary matrices). A middle ground could have been ResNet :) Let us know if you have further questions.
>
> > **TFs are not trained to do CoT:** We would like to remark that the next token prediction loss exactly mimics the CoT-I structure. What is unclear is whether the text in the pretraining data has a compositional structure. We believe that Nye et al. (check ref)'s work on "take it step by step" actually indicates that this is true since the pretrained model responds to the "step by step" prompt.
>
> > **Ablation experiments (attached Fig 4):** In attached Figure 4, we run the ICL experiment based on the settings the reviewer suggests. Due to time limitations, we are still working on the scenario where intermediate space is filled with repeatting data. Here, we train a small GPT-2 with $d=10$ and $k=8$. Blue, orange, and green curves show the evaluation results using the methods introduced in the paper. Here, we also try a variant setting where the prompt is provided in the form $(x,[?],y)$, and solid, dashed black curves show the results where the padded intermediate feature [?] is random and constant $-1$, respectively. We choose $-1$ as the constant number since the correct feature is outputed after ReLU and is non-negative. Results have shown that without given meaningfully intermediate information, ICL finds it hard to target the correct task functions.
>
> Finally, we apologize for missing the limitations discussion, and we agree with the reviewer that this work lacks an explanation of the connection between our MLP-based setting and LLM applications. We will further modify our paper to articulate our work more deeply and clearly.

---

> > ### Comment · Reviewer_zLZS · 2023-08-10
> > **Thank you!**
> >
> > Thank you for your clarifications and running the additional experiments.
> > Given the commitment for restructuring the paper, I vote to accept the paper but will not increase my score further. As a few of the reviewers also had troubles understanding the paper, I feel that this was indeed a shortcoming of the paper that hopefully will be resolved if accepted. I encourage the other reviewers to rethink their rather, in my opinion, low scores given that the authors will invest time to improve the papers presentation. I find the results very valuable for the audience of NeurIPS and should be discussed at the conference.

---

> > > ### Author Response · Authors · 2023-08-13
> > >
> > > We thank the reviewer for the encouraging assessment and positive recommendations. Their suggestions definitely helped improve the organization and clarity of our paper.

---

### Official Review · Reviewer_8qtM · 2023-07-06

**Soundness:** 3 good
**Presentation:** 3 good
**Contribution:** 3 good
**Rating:** 5
**Confidence:** 4

**Summary:**

This paper aims to demystify the mechanism lying in the in-context learning (ICL) and chain-of-thought (CoT). It reveals how CoT significantly reduces the sample complexity of ICL. It uses a two-layer MLP, and a backbone GPT-2 model for exploration.   The experimental results reveal some interesting findings, e.g., the in-context samples needed is linearly dependent on the input dimension. The paper also provides theoretical analysis of probable approximation of MLPs via chain-of-thought.

**Strengths:**

- The paper provides both experimental and theoretical evidence to support its claims about how CoT works.
- The findings about the relations between the in-context samples needed and the input MLP dimension are interesting and insightful.

**Weaknesses:**

- Lack of clarity: Some parts of the paper may be unclear or difficult to follow, which could make it challenging for readers to understand the key findings and contributions of the study. Some of the concepts are not explained very well in the beginning (e.g, what is compositional learning? What is MLPs in-context), thus making the paper abstruse.
- The main contribution of the paper is not clear. The paper claims to dissect the mechanism of CoT

**Questions:**

- How do you use the intermediate MLPs features as the CoT prompts for the GPT-2? It seems that the features are not tokens to be prompts.
- Why do you use the input dimension d and hidden size k of a MLP layer as the measures for experiments? Are there any specific reasons to use them as the proxy for some properties of the target task?

**Limitations:**

No limitation section is found.

---

> ### Author Rebuttal · Authors · 2023-08-09
>
> We thank the reviewer for their careful reading and for recognizing the interest and insights in our work. We hope that we have addressed your questions and concerns adequately below.
>
> > **Lack of clarity:** We apologize to the reviewer for any confusion stemming from the unclear aspects of our work. Our commitment to enhancing clarity will be reflected in our planned revisions. While we're unable to make changes during this review stage, our general response outlines our reorganization strategy, and we're eager to consider any additional recommendations. We'll also refine the introduction to better articulate our motivations, main contributions, and notations.
> > * **_Compositional learning_:** It targets a learning method that decomposes a complex problem into several intermediate stages, addressing each sequentially and combining the solutions to render a final prediction. CoT illustrates this method well. Unlike ICL, which requires recovering a function $f$ from data such as $(x_1,y_1,x_2,y_2,...)$ where $y=f(x)$, CoT utilizes prompts of the form $(x_1,s_1,y_1,x_2,s_2,y_2,...)$ where $s=g_1(x)$ and $y=g_2(s)$, and solves the functions $g_1,g_2$ separately. By composing them to obtain $f:=g_2\circ g_1$, CoT leverages a compositional advantage. This makes the process of solving subfunctions more feasible and sample efficient.
> > * **_MLPs in-context_:** We regret the ambiguity. Our statement that "_transformers can in-context learn MLPs_" means that transformers are capable of learning MLP tasks from in-context samples without altering the model weights, which diverges from traditional learning scenarios where model weights are tuned to optimize performance on specific tasks.
>
> > **Contribution is not clear:** We apologize that this does not come through. In this work, our primary aim is to dissect and elucidate the mechanism of CoT in a more accessible and simplified setting. In the context of LLMs, the complexity arising from the vast amount of pretraining data and intricate semantic information makes it challenging to discern why concepts such as "let's think step by step" are effective. Our work confronts this challenge by modeling the CoT prompt using random MLPs. Through both empirical and theoretical results, we demonstrate that the transformer's strength in CoT lies in its ability to learn compositional functions and separate out filtering and learning processes. This novel understanding of CoT's underlying mechanism results in significantly improved sample efficiency in both the training and in-context inference stages. We have included an attached Figure 1 to offer a visual representation of these findings, enhancing understanding, and have also provided a more comprehensive explanation in the general response.
>
> > **Tokens vs prompts:** We concur that GPT-2 contains embedding layer that embeds language words into continuous vector. For our MLP-based prompting study, we clarify that we've substituted the GPT-2 embedding layer with up and down linear projections, aligning with Garg et al. This allows tokens to map into the same dimension, and we'll add further implementation details in our experiment section.
>
> > **Why use $d$ and $k$ as proxy for target task difficulty?** We thank the reviewer for this insightful question. To study the compositional learning ability and sample complexity of CoT, in this work, we explore the learning of complex functions $f=g_2\circ g_1$, which can be decomposed into two less intricate tasks. Using 2-layer MLPs as examples, the intermediate feature is generated via $s=g_1(x):=\phi(Wx)$ with $W\in\mathbb{R}^{k\times d}$ presenting the weights of the first layer and $\phi$ as the activation. The second/final layer is functionalized by $y=g_2(s):=v^\top s$ where $v\in\mathbb{R}^k$. CoT prompts are thus framed as $(x,s,y)$. Based on this setting, $(d,k)$ are the hyperparameters that control the complexity of the MLP, and increasing $d$ and $k$ correspondingly heightens the difficulty of recovering the MLP. To solve 2-layer MLP with dimensions $d,k$, ICL indeed necessitates $O(d*k)$ in-context samples, whereas CoT only requires $O(\max(d,k))$ samples, thanks to its innate compositional learning capacity.
>
> We sincerely apologize for failing to include the limitations in the current version of our paper. We acknowledge that addressing the potential limitations is vital for a complete understanding of our work, and we have taken steps to include them in our general response. We are also committed to incorporating the reviewer's insightful comments to enhance our paper by adding further clarification and details to highlight and motivate our primary contributions. Thank you for bringing these vital aspects to our attention.

---

> > ### Comment · Reviewer_8qtM · 2023-08-20
> >
> > Thanks for the response. I don't have any further questions.

---

### Official Review · Reviewer_mo8c · 2023-07-21

**Soundness:** 2 fair
**Presentation:** 1 poor
**Contribution:** 2 fair
**Rating:** 5
**Confidence:** 3

**Summary:**

In this paper, the authors explore the mechanics of Chain-of-Thought (CoT), a method that has successfully enabled language models to handle complex reasoning tasks by decomposing them into simpler steps.
The study aims to understand the underlying mechanics of CoT by investigating its impact on the ability of transformers to in-context learn a simple yet general family of compositional functions: multi-layer perceptrons (MLPs).
The authors reveal that the success of CoT can be attributed to two distinct phases: focusing on data related to each step of the composition and in-context learning the single-step composition function.
They provide experimental and theoretical evidence that demonstrates how CoT significantly reduces the sample complexity of in-context learning (ICL) and facilitates the learning of complex functions that non-CoT methods struggle with.

**Strengths:**

1. This work explores CoT in the learning of MLPs thatis a novel perspective. Such a simplicity can also help to dig the into inner machenism of CoT and ICL.
2. This work experimentally compare three schemes (ICL, CoT-I, and CoT-I/O), providing valuable insights into their differences and the benefits of CoT prompting.
3. The paper provides a formalized theorem that explains how a transformer architecture can realize the CoT process for MLPs.

**Weaknesses:**

1. The Chain-of-Thought (CoT) implementation presented in this study appears to be limited in terms of reasoning steps, which may not be fully consistent with the original motivation behind proposing CoT as a method for complex reasoning tasks.

In most existing work, CoT has the potential to assist in various complex reasoning tasks as it provides the model with step-by-step guidance on solving the input.
Each reasoning step should include both the intermediate state and the process through which this state is generated from the previous one.
This is typically implemented using natural language descriptions or formal language expressions.

However, the CoT implementation in this paper focuses on the intermediate states in MLPs and does not provide information on how each intermediate state is produced.
By only presenting intermediate states rather than complete reasoning steps, the CoT explored in this study primarily reflects the capability of state transition rather than reasoning.

2. The study appears to have limited analysis on generalization capabilities, which is a fundamental aspect of in-context learning (ICL).

This study assumes that there is no distribution shift between the training and test datasets.
However, a fundamental aspect of in-context learning (ICL) is the ability to learn new tasks from in-context examples.
Consequently, it is crucial to examine the performance in a generalization setting.
For example, [1], which shares a similar philosophy with this work, extensively discussed the behavior of ICL on out-of-distribution prompts.

[1] Shivam Garg, Dimitris Tsipras, Percy S Liang, and Gregory Valiant. What can transformers learn in-context? a case study of simple function classes.


**Questions:**

1. Do the conclusions drawn in this study remain consistent in an out-of-distribution setting? For example, what might occur if the MLPs in the test and training sets exhibit notable differences, such as having different widths?

**Limitations:**

This paper does not explicitly address its limitations. The most significant limitation of the study is the unclear contribution towards answering the question of how the Chain-of-Thought (CoT) method can assist in solving complex reasoning tasks.

---

> ### Author Rebuttal · Authors · 2023-08-09
>
> We thank the reviewer for their detailed feedback and for recognizing the novelty and value our results contribute, particularly in understanding the inner mechanisms of CoT and ICL.
>
> >**Reasoning steps vs intermediate state:** We appreciate the reviewer's insightful comment on this matter. In the realm of NLP, the reasoning process targeting the intermediate state is intricately interwoven within the language sentences and pretrained LLMs. Our work, however, highlights that the advantage of CoT over ICL lies in the decomposition of complex problems into intermediate steps, rather than merely its competence in semantic comprehension. Though our MLP setting may seem to lack a distinct "reasoning process", it does implicitly include reasoning within positional embedding. Inspired by prior work [Garg et al. 2022, Wei et al. 2022], we employ a CoT-based MLP setting, enabling us to precisely and distinctly examine the underlying CoT mechanism.
> >
> >We respectfully disagree with the claim that "_the CoT explored in this study primarily reflects the capability of state transition rather than reasoning_". The process of ICL/CoT involves acquiring the ability to perform a novel task through demonstrations. When provided with too few in-context examples, accurate predictions become unattainable, as evidenced by the figures in our paper where fewer samples (smaller x-axis) correspond to larger errors. Thus, CoT method involves not merely state transition but an in-context learning phase to discern and restore the feature generation function.
>
> > **Out-of-distribution evaluations (Fig 2 in the attached file):**  Great point! Our investigation has focused on the standard setting where train and test samples have identical distribution. Regrettably, we did not explore the out-of-distribution aspect of CoT in our original manuscript. Following the reviewer's suggestion, we have commenced new experiments, with preliminary findings shown in the attached Figure 2. In short, **our observations on CoT translate well to several out-of-distribution setting such as incorporating label or feature noise and misspecified input or hidden dimensions**. To elucidate our results and examine the test risk under varying distribution shift levels, we plot test risk evaluated when prompt has 100 in-context examples. In Fig 2(a), we analyze noisy in-context samples during testing. The solid and dashed curves represent the test risks, corresponding to the noisy in-context samples whose (input, output) takes the form of either $(x,y+\text{noise})$ or $(x+\text{noise},y)$, respectively. The results indicate that CoT exhibits greater robustness compared to ICL, and the test risks increase linearly with the noise level, with attributed to the randomized MLPs setting. Additionally, in Fig 2(b)(c), we instead explore out-of-distribution test tasks where test MLPs differ in $(d,k)$ from the training phase. For both subfigures, we firstly train small GPT-2 using 2-layer MLPs with $d=10,k=8$. In Fig 2(b), we fix $d=10$ and vary $k$ from $1$ to $8$, whereas in Fig 2(c), we fix $k=8$ and vary $d$ from $1$ to $10$.  In both instances, the findings reveal that CoT's performance remains almost consistent when $k\geq 4$ or $d\geq6$, and ICL is unable to surpass it. The improved performance of ICL with smaller values of $d$ or $k$ again reinforces our central assertion that ICL requires $O(d*k)$ samples for in-context learning of the 2-layer random MLP, and reducing either $k$ or $d$ helps in improving the performance. Given that we employ the ReLU activation function, smaller values of $d$ or $f$ can lead to significant bias in the intermediate feature. Consequently, CoT cannot derive substantial benefits from this scenario, resulting in a decline in performance. To sum up, we thank the reviewer for raising this issue and will integrate OOD experiments in our revision.
>
> > **Robustness implications of our work on broader in-context learning** We also briefly discuss how our formalism that decouples CoT into filtering and in-context learning stages has adversarial/distributional robustness implications for in-context learning. The related works on in-context learning such as [Garg et al. NeurIPS'21, Akyurek et al. ICLR'23] focus on the scenario where input prompt contains IID (input, label) pairs. A natural question is: **What if the prompt contains non-IID data, for instance, is in-context learning robust to outlier features?** In this work, we show that transformer can provably *filter a heterogeneous prompt* to obtain an *purified prompt containing IID features* amenable to in-context learning (our Theorem 1). Thus, beyond CoT, our work has implications for outlier/adversarially-robust in-context learning. Our Filtering+ICL formalism has the following interpretation: *Train the transformer with heterogeneous prompts so that it learns how-to-filter. Then, it can implement outlier-robust ICL during inference time*. We will add a discussion on this in the last section.
>
> > **Lack of limitations:** We acknowledge that we should have provided an explicit discussion of limitations. We've now addressed potential limitations in our general response and intend to include a designated subsection in our revised version.
>
> We extend our gratitude to the reviewer once more for their insightful comments. In response to their feedback, we are actively engaged in enhancing our discussion to elucidate our problem more comprehensively. Additionally, we are delving deeper into the out-of-distribution aspect by conducting further experiments with varying parameters, such as different $(d,k)$ MLPs pretrained models, alternative GPT-2 architectures, diverse types of distribution, and more.

---

> > ### Comment · Reviewer_mo8c · 2023-08-16
> > **Thank You!**
> >
> > I would like to raise the score from 4 to 5, as the additional experiments addressed my concerns regarding generalization. Nevertheless, I still do not comprehend how this study contributes to elucidating the mechanism of CoT in practical reasoning tasks. Consequently, I can only assign a borderline rating for this work.

---

> > > ### Author Response · Authors · 2023-08-17
> > > **Thanks and further clarification on CoT**
> > >
> > > Dear Reviewer,
> > >
> > > Thank you for your experiment suggestions and for reevaluating our work. We acknowledge your concern. Below, we discuss the core features of CoT, based on which, we will explicitly distinguish **few-shot CoT** and **zero-shot CoT** in the final manuscript. We hope that this can address some of your concerns.
> > >
> > >  1. **Benefits of step-by-step problem solving:** Core of CoT is decomposing complex tasks and our work distills this essence into the MLP setting. Although we agree that practical CoT is not as structured as our MLP prompt, we establish **clear theoretical and empirical benefits of "step-by-step problem solving"** in terms of sample efficiency as well as model expressivity.
> > >
> > >  2. **The strategy CoT uses to solve the problem:** In practice, CoT (step-by-step decomposition) can happen in two ways:
> > >
> > > - **Option 1: Few-shot CoT.** In this setting, transformer leverages the examples and associated solutions provided in the context window to solve the new problem. For instance, it solves a new math problem by studying related problems and their solutions in-context. Most of our results, namely two-layer MLPs, and theory focus on this **few-shot** setting. Here, transformer infers "state transitions" (i.e. weight matrices of MLP) from in-context examples.
> > >
> > > - **Option 2: Zero-shot CoT.** In this setting, transformer creates the solution steps without any relevant examples. An intuitive explanation is that the model maps the input problem to a **memorized set of "skills" and skill transitions** and apply them step-by-step. For instance, standard operations like "+,-,x,/" can be memorized during pretraining. Zero-shot is possible because these memorized "skills" and "transitions" does not have to be inferred in-context (unlike Option 1). Our **Section 4.3** provides insights into this via Deep Linear MLPs. Here, unlike 2-layer MLPs with continuous weights, we use finitely many weight matrices, thus, transitions are discrete. Each matrix correspond to a "skill" that can be memorized by the transformer during training. Confirming this intuition, **Figure 6(a)** demonstrates that chain-of-thought can succeed with a single example whereas ICL needs more. This is because, with CoT, transformer can memorize the $K=4$ matrices and compose them, whereas ICL can't memorize all $K^L=4^6$ variations. Here, the single example reminds the transformer which skills to select from the memorized repertoire. This would in-fact work zero-shot if the skills can be determined from the input features, specifically, when input features serve as an informative "initial state" to kickstart the skill chain. However, as the reviewer has also mentioned, we have not delved into the informativeness of input features, and instead focused on learning state transitions.
> > >
> > > In the final manuscript, we will clearly distinguish few-shot CoT vs zero-shot CoT and provide a discussion of limitations, e.g., zero-shot inference of the chain from the input features.
> > >
> > > Thanks again for your valuable insights and we would be grateful to hear further feedback,
> > >
> > > Authors

---

### Author Rebuttal · Authors · 2023-08-10

We thank the reviewers for their constructive comments and insightful questions. We are gratified that many of the reviewers found our work insightful and interesting. In the following sections, we will summarize our key contributions, respond to shared concerns raised by the reviewers, and provide explanations for the new figures presented.

We briefly recap our **main contribution and message**: There are several major works on in-context learning (ICL) theory (e.g. [Xie et al. ICLR'22, Garg et al. NeurIPS'22]). These focus on learning from the examples provided in the prompt. Compared to these, we consider a novel setting where prompt contains chain-of-features obtained from a multistep problem-solving (or reasoning) process. We show that when solving a new problem, **chain-of-thought operates by (1) filtering the heterogeneous prompt and retrieving the relevant features, (2) running in-context learning on the filtered prompt, and (3) Loop back to (1).** This process is illustrated in attached Figure 1. Our key contributions are establishing how transformers can provably implement this process and demonstrating the empirical and theoretical value of CoT in terms of approximation and sample complexity. In our Theorem 1, the attention mechanism provides a critical role in both filtering and ICL stages highlighting the transformer-specific nature of our results. We will further elaborate on this in the final manuscript.

To proceed, we address the shared concerns by the reviewers:

- **Paper Organization.** Reviewers 8qtM, zLZS, and wZge noted concerns with the paper's clarity, citing confusion over the existence of both empirical and experimental results sections. We appreciate their feedback and have laid out a plan to enhance the paper's readability:
    1. Section 3 will be focused solely on theory, and additional discussion will be included to better motivate and clarify our theoretical results. For instance, we will include an illustration to elucidate how the theory functions (see Fig. 1 in the .pdf and also the response to Reviewer wZge for further details).
    2. All experiments involving 2-layer MLPs will be consolidated in Section 4.
    3. Section 5 will address the training aspects of CoT by exploring deep linear MLPs.

- **Lack of Limitations Section.** We recognize our oversight in neglecting to include an explicit discussion of the paper's limitations and sincerely apologize for this omission. Rest assured, we will incorporate a limitations section in our revised manuscript. Our notable limitations include:
    1. We should have provided a deeper study of out-of-distribution scenarios. In response to Reviewer mo8c, we have partly addressed this limitation (see Fig 2 in the pdf).
    2. Our key contributions are theoretical and we work with a synthetic problem setting which extends [Garg et al. NeurIPS'22, Akyurek et al. ICLR'23] to compositional functions (namely MLPs) and CoT.

**Supporting experiments:** We provide the following figures to address reviewer inquiries.

- Figure 1 aims to clarify the chain-of-thought prompting formalized by our work by visualizing how CoT can be decoupled into filtering and in-context-learning stages.
- Figure 2 provides new robustness experiments in response to Reviewer mo8c. In Fig 2(a), we add noise to the label $y$ (solid) or input features $x$ (dashed). This shows that CoT methods (trained on noiseless data) are fairly robust to noisy data at test-time. In Fig 2(b), we in-context learn an MLP with $k$ hidden nodes whereas transformer is trained for MLPs with $8$ hidden nodes. This shows that CoT-I/O is robust to misspecification as long as it is small. However, if $k$ is very small, then CoT-I/O suffers more distribution shift. This makes sense because CoT-I/O relies more on hidden features compared to others. In Fig 2(c), we consider misspecification of input dimension: TF is trained with $d=10$ but we feed a neural net with $d<10$. This reveals that CoT is pretty robust and, in fact, CoT-I is the most robust to misspecification. We speculate this is because CoT-I makes similar use of input and hidden features.
- In Figure 3, we investigate deep nonlinear MLPs. The main takeaway is that, more CoT steps help but not that much. We believe this might be because of the random generation of neural nets. Specifically, it is possible that when neural net weights are fully random, a two layer neural net might be able to accurately approximate a 4 layer network. This would mean only 2 CoT steps are needed to do a good job.
- In Figure 4, we put vanilla ICL in CoT format and feed $(x,[?],y)$ in response to Reviewer zLZS. We set [?] mark as random gaussian features or all [-1] vector. In both cases, the performance coincide with vanilla ICL $(x,y)$ which we found to be intuitive.
- In Figure 5, we train the models with more in-context examples as requested by Reviewer wZge. The main conclusion is that: For small GPT, ICL can indeed not approximate the neural net even with many examples (unlike CoT) whereas, for large GPT, ICL can do so (although much less efficient). This is in line with our theoretical intuitions on the expressivity benefits of CoT.


Finally, as for the concerns/questions raised, we believe that we successfully addressed all of them sufficiently and reply in line to each review. We would be grateful to respond further reviewer inquiries during the discussion week.

---

### Decision · Program_Chairs · 2023-09-21

**Decision:**

Accept (poster)

**Comment:**

The authors make a novel contribution towards understanding the phenomenon of CoT. Reviewers converged on recommending acceptance after a lively rebuttal phase. One lingering concern is about the clarity of presentation -- but much of this is likely due to the abstract nature of the contribution and the authors have made a strong effort to respond to reviewer feedback during rebuttal + laid out a plan for improving it. The contribution is valuable and novel, and worth discussion at NeurIPS